

# Formation and impacts of nitryl chloride in Pearl River Delta

Haichao Wang[1,4], Bin Yuan[2,3,*], E Zheng[2,3], Xiaoxiao Zhang[2,3], Jie Wang[1], Keding Lu[5,6], Chenshuo Ye[2,3],

Lei Yang[2,3], Shan Huang[2,3], Weiwei Hu[7], Suxia Yang[2,3], Yuwen Peng[2,3], Jipeng Qi[2,3], Sihang Wang[2,3],

Xianjun He[2,3], Yubin Chen[2,3], Tiange Li[2,3], Wenjie Wang[2,8], Yibo Huangfu[2,3], Xiaobing Li[2,3], Mingfu

Cai[2,3], Xuemei Wang[2,3], Min Shao[2,3]

[1] School of Atmospheric Sciences, Sun Yat-sen University, Zhuhai, 519082, China

[2] Institute for Environmental and Climate Research, Jinan University, Guangzhou 511443, China

[3] Guangdong–Hong Kong–Macau Joint Laboratory of Collaborative Innovation for Environmental

Quality, Guangzhou, 511443, China

[4] Guangdong Provincial Observation and Research Station for Climate Environment and Air Quality

Change in the Pearl River Estuary, Key Laboratory of Tropical Atmosphere-Ocean System, Ministry

of Education, Southern Marine Science and Engineering Guangdong Laboratory (Zhuhai), Zhuhai,

519082, China

[5] State Key Joint Laboratory of Environmental Simulation and Pollution Control, College of

Environmental Sciences and Engineering, Peking University, Beijing, 100871, China.

[6] The State Environmental Protection Key Laboratory of Atmospheric Ozone Pollution Control,

College of Environmental Sciences and Engineering, Peking University, Beijing, 100871, China

[7] State Key Laboratory of Organic Geochemistry and Guangdong Key Laboratory of Environmental

Protection and Resources Utilization, Guangzhou Institute of Geochemistry, Chinese Academy of

Sciences, Guangzhou 510640, China

[8] Multiphase Chemistry Department, Max Planck Institute for Chemistry, Mainz 55128, Germany

Correspondence: Bin Yuan (byuan@jnu.edu.cn)





**Abstract.** Here we present a field measurement of $ClNO_2$ (nitryl chloride) and $N_2O_5$ (dinitrogen
pentoxide) by a Time-of-Flight Chemical Ionization Mass Spectrometer (ToF-CIMS) with the Filter
Inlet for Gas and AEROsols (FIGAERO) at a regional site in Pearl River Delta during a photochemical
pollution season from Sept. 26th to Nov. 17th, 2019. Three patterns of air masses are sampled during
this campaign, including the dominating air masses from north and northeast urban regions (Type A),
the southeast coast (Type B) and the South China Sea (Type C). The concentration of $ClNO_2$ and $N_2O_5$
were observed much higher in Type A and B than those in Type C, indicated the urban nighttime
chemistry is more active than the background marine regions. $N_2O_5$ uptake coefficient and $ClNO_2$
production yield were estimated by measured parameters, and the performance of the previously
derived parameterizations were assessed. The nighttime $ClNO_2$ correlated with particulate chloride
and the mass concentration of fine particles (most likely due to aerosol surface area), but not with
nitrate radical formation rate, suggested the $ClNO_2$ formation was limited by the $N_2O_5$ uptake rather
than $N_2O_5$ source at this site. By examining the relationship of particulate chloride and other species,
we implied that anthropogenic emissions (e.g., biomass burning) rather than sea salt particles dominate
the origin of particulate chloride, despite the site is only about 100 km away from the ocean. A box
model with detailed chloride chemistry is used to investigate the impacts of $ClNO_2$ chemistry on
atmospheric oxidation. Model simulations showed the chloride radical liberated by $ClNO_2$ photolysis
during the next day had a small increase in concentrations of OH, $HO_2$ and $RO_2$ radicals, as well as
minor contributions to $RO_2$ radical and $O_3$ formation (<5%, on daytime average) in all the three types
of air masses. Relative higher contributions were observed in Type A and B. The overall low
contributions of $ClNO_2$ to atmospheric oxidation are consistent with those reported recently from
wintertime observations in China (included Shanghai, Beijing, Wangdu and Mt. Tai). This may be
attributed to: (1) Relative low particle mass concentration limited $ClNO_2$ formation; (2) Other reactions
channels, like nitrous acid (HONO), oxygenated volatile organic compounds (OVOCs, including
formaldehyde) and ozone photolysis, had larger radical formation rate during the ozone pollution
episodes and weakened the $ClNO_2$ contribution indirectly. The results provided scientific insights into
the role of nighttime chemistry in photochemical pollution under various scenarios in coastal areas.





**1. Introduction**


Chloride radical is an important oxidant in the tropospheric besides OH radicals, $NO_3$ radicals and
ozone (Saiz-Lopez and von Glasow, 2012;Simpson et al., 2015;Wang et al., 2019b), which alters the
fate of many atmospheric compositions including oxidants, reactive nitrogen compounds, volatile
organic compounds (VOCs), and other halogens. Cl radical is much more reactive than OH with
respect to certain VOCs (e.g., alkanes) by a few orders of magnitude for reaction rate constant
(Atkinson and Arey, 2003;Atkinson et al., 2006), therefore, it contributes to atmospheric oxidation
capacity considerably in the troposphere despite low concentrations. For example, global model
showed about 20 % of ethane, 14 % of propane oxidation are attributed to the chloride chemistry at
the global scale (Wang et al., 2019c). Modeling simulations also demonstrated chloride chemistry
enhanced oxidative degradation of VOCs by >20% at some locations (Sarwar et al., 2014).
Photolysis of $ClNO_2$ (R1) is a major source of the tropospheric chloride radical (Thornton et al.,
2010b;Simpson et al., 2015), other chloride radical sources include the reaction of HCl with OH
(Riedel et al., 2012;Eger et al., 2019), photolysis of $Cl_2$ and other halogen compounds like ICl and
BrCl (Peng et al., 2021). Tropospheric $ClNO_2$ is not only an important chlorine activation precursor
but also a nocturnal resouvior of reactive nitrogen, which is mainly formed in heterogeneous reaction
of $N_2O_5$ on chlorine-containing particles with a branch ratio at nighttime (R2).
$$ClNO_2 + h\nu \;\rightarrow\; Cl + NO_2 \qquad\qquad (R1)$$
$$N_2O_5 + H_2O/Cl^- \;\rightarrow\; \varphi ClNO_2 + (2-\varphi)NO_3^- \qquad (R2)$$
where $\varphi$ represents the yield of $ClNO_2$. This mechanism was firstly proposed by Finlaysonpitts et al.
(1989) through detecting the products of $N_2O_5$ uptake on NaCl particles. Given this reaction, the
formation of $ClNO_2$ can be influenced by the $N_2O_5$ uptake (such as $N_2O_5$ uptake probabilities and
aerosol surface area) as well as the production yield of $ClNO_2$.
$N_2O_5$ uptake coefficient, $\gamma(N_2O_5)$, have been reported highly varied under tropospheric conditions
(Brown and Stutz, 2012). Both the field and laboratory studies revealed that this process can be affected
by ambient temperature, relative humidity (Mozurkewich and Calvert, 1988;Mentel et al.,
1999;Hallquist et al., 2003), chemical compositions (such as the content of nitrate, liquid water,
chloride, and organics) (Mentel et al., 1999;Brown et al., 2006;Bertram and Thornton, 2009;Gaston et
al., 2014;McDuffie et al., 2018b;Tang et al., 2014;Anttila et al., 2006), as well as particle morphology
(Mielke et al., 2013;Zong et al., 2021). Until now, the key factors that controlling $N_2O_5$ uptake





coefficient in the different environments are still not well understood. $ClNO_2$ yield is also highly varied
subject to the liquid water and chloride content in the aerosol (Behnke et al., 1997;Roberts et al.,
2009;Bertram and Thornton, 2009). Several studies demonstrated that the $ClNO_2$ yield is also affected
by other factors like aerosol sulfate (Staudt et al., 2019) and organics (Ryder et al., 2015;Tham et al.,
2018;McDuffie et al., 2018a). However, the comprehensive quantitative relationship of these factors
in controlling the yield still has large uncertainties. These gaps in understanding the critical controlling
factors for $N_2O_5$ uptake coefficient as well as $ClNO_2$ yield lead to the prediction of $ClNO_2$ and
particulate nitrate production very challenge.
Osthoff et al. (2008) and Thornton et al. (2010a) directly observed elevated $ClNO_2$ in coastal and
inland U.S. by chemical ionization mass spectrometer (CIMS), respecitively. They shed light on the
significance of $ClNO_2$ photolysis in launching the radical chemistry during the morning time, and also
affecting halogen chemistry and reactive nitrogen cycling. Large amounts of chloride radicals are
liberated through the photolysis of noctural accumulated $ClNO_2$ (R1), which oxidizes VOCs and
produces peroxy radicals ($RO_2$) to initiate the daytime raidcal cycling in the morning, when other
radical source, like ozonolysis and photolysis of $O_3$, HONO and HCHO, are still weak (Osthoff et al.,
2008). The impacts of $ClNO_2$ chemistry on primary source of radicals and ozone formation is a critical
topic, the answer of which is very helpful to narrow the gap of the missing priamry source of ROx and
improve our knowledge of the currect ozone pollution mechanism (Tan et al., 2017;Tham et al., 2016).
Model simulation highlighted $ClNO_2$ chemistry could increase mean daily maximum 8 h ozone by up
to 7.0 ppbv in some areas in the Northern Hemisphere (Sarwar et al., 2014). The large contribution
was also confirmed in the southern California region by a box model study (Riedel et al., 2014). In
addition, global model simualtion showed $ClNO_2$ chemistry increases wintertime ozone by up to 8 ppb
over polluted continents (Wang et al., 2019c). Particularly, previously modelling results also highlight
the importance of $ClNO_2$ chemistry in enhancing $O_3$ production in China (Li et al., 2016;Yang et al.,
2022b).
Several field studies reported the measurement of $ClNO_2$ in varied environments in the past decade
(Riedel et al., 2012;Young et al., 2012;Mielke et al., 2013;Riedel et al., 2013;Bannan et al., 2015;Faxon
et al., 2015;Mielke et al., 2015;Phillips et al., 2016;Bannan et al., 2017;Wang et al., 2017c;Wang et al.,
2017d;Le Breton et al., 2018;McDuffie et al., 2018a;Yun et al., 2018a;Zhou et al., 2018;Bannan et al.,
2019;Eger et al., 2019;Haskins et al., 2019;Jeong et al., 2019;Xia et al., 2020;Xia et al., 2021;Tham et
al., 2016;Tham et al., 2014;Wang et al., 2016;Phillips et al., 2012;Lou et al., 2022;Sommariva et al.,
2018), in which the maximum $ClNO_2$ up to sub-ppbv to several ppbv were reported, indicating its
ubiquity presence worldwide and a broad atmospheric impacts over various regions. During the
CalNex-LA campaign 2010, $ClNO_2$ was measured at ground site, the Research Vessel and aircraft



platform, which depicted a full picture of the abundance of $ClNO_2$ and confirmed its large impacts on
atmospheric chemsitry in both urban and coastal regions in California (Riedel et al., 2012;Young et al.,
2012;Mielke et al., 2013). Recently, Wang et al. (2016) used a box model simulated the chemical
evolution of the plume after leaving the observation site in Hongkong and showed $ClNO_2$ chemistry
had a following-day enhancement of ozone peak and daytime ozone production rate by 5–16% and
11–41%, along with a large increasing of OH, $HO_2$ and $RO_2$ concentration especially in the morning.
While Xia et al. (2021) and Lou et al. (2022) reported winter measurments of $ClNO_2$ in north and east
China, respectively, both they showed moderate $ClNO_2$ level and a relative small contributions of
$ClNO_2$ chemistry to radical source and ozone enhancement on campaign average. These results is quite
different with that happened during the summertime in China (Tham et al., 2016;Wang et al., 2016;Tan
et al., 2017), and highlight the large variation of $ClNO_2$ chemistry influenced by temporal spatial
distribution.

127        Despite its likely importance to the regional atmospheric oxidation and air quality, investigations of

$ClNO_2$ chemistry in China remain relatively sparse. There are several field measurements of $ClNO_2$
conducted in the China in recent years, while considering the large diversities of air mass in inland and
coastal regions in China, more field and model works are need to gain more insights to the $ClNO_2$
chemistry in various atmospheric environments and assess its atmospheric impacts. Until now, only
several field measurement of $ClNO_2$ were reported in Pearl River Delta (PRD) region (Tham et al.,
2014;Wang et al., 2016;Yun et al., 2018a), and only Wang et al. (2016) reported a comprehensive
analysis of the impact of $ClNO_2$ chemistry on radical and ozone formation in 2013 as mentioned before.
To understanding the increasing $O_3$ problem in recent years (Wang et al., 2019a) and examining the
role of $ClNO_2$ chemistry in $O_3$ formation in PRD, we measured $ClNO_2$, $N_2O_5$, and other related
parameters at a regional site in PRD during a severe photochemical pollution season in 2019. The
abundance, formation, and variation during different air masses patterns are well characterized. The
factors impact its formation are diagnosed. Finally, the contribution of chloride radicals liberated by
$ClNO_2$ photolysis on the daytime radical chemistry, as well as ozone formation are comprehensively
assessed by a box model coupled with detailed chloride chemistry.

## 2. Method

### 2.1 Measurement site

This campaign was conducted at the Guangdong Atmospheric Supersite of China, which is located on
the top of a mountain (~ 60 m high) in Heshan (22.728°N, 112.929°E), Jiangmen city, Guangdong
Province (Yang et al., 2022a) . This site was in the western Peral River Delta where no major industry
in the surroundings, but with some farmland and a few residents live at the hill foot. The traffic is far



away from this site and believed seldom disturbs the sampling. The anthropogenic activity is much
lower than the urban regions like Guangzhou City, but the air quality is often influenced by neighbor
cities, especially the outflow of air masses from the regions on the north and northeast. Therefore, the
air masses sampled at this site are representative of the urban pollution from the center PRD. There
were many atmospheric intensive studies once conducted in the site to study the air pollutions in PRD
(Tan et al., 2019;Yun et al., 2018b). In this study, the instruments were located on the top floor of the
measurement building with inlets approximately 15 m above the ground. The data presented in the
study were collected from 27$^{th}$ September to 17$^{th}$ November 2019, during which photochemical
pollution occurred frequently (Yang et al., 2022a). Time is given as CNST (Chinese National Standard
Time = UTC+8 h). During the campaign, sunrise was at 06:00 and sunset was at 18:00 CNST.
**2.2 Instrument setup**
A comprehensive suite of instrumentation was overviewed and listed in Table 1. An iodide-adduct
Time-of-Flight Chemical Ionization Mass Spectrometer (ToF-CIMS) with the Filter Inlet for Gas and
AEROsols (FIGAERO) was applied to measure $ClNO_2$ and $N_2O_5$ along with other oxygenated organic
species (Ye et al., 2021;Wang et al., 2020b) . In brief, the gas phase species were measured via a 2-m-
long, 6-mm-outer-diameter PFA inlet while the particles were simultaneously collected on a Teflon
filter via a separate 2-m-long, 10-mm-outer-diameter copper tubing inlet; both had flow rates of 2 L
min$^{-1}$ with a drainage flow of 20 L min$^{-1}$. The gas phase was measured for 25 minutes at 1 Hz, and the
FIGAERO instrument was then switched to place the filter in front of the ion molecule region; it was
then heated incrementally to 200 °C to desorb all the mass from the filter to be measured in the gas
phase, which resulted in high-resolution thermograms. $ClNO_2$ and $N_2O_5$ are measured as the iodide
adduct ions at m/z 207.867 ($IClNO_2^-$) and m/z 234.886 ($IN_2O_5^-$) in the ToF-CIMS, respectively. The
sensitivities for detecting $ClNO_2$ and $N_2O_5$ with the dependence of water content were quantified and
described in details (see Appendix). The limit of detection (LOD) for $ClNO_2$ and $N_2O_5$ were 4.3 and
6.0 pptv in 1-minute time-resolution, respectively, with an uncertainty of ~30%.
Sub-micron aerosol composition ($PM_1$) were measured by a High-Resolution Time of Flight Aerosol
Mass Spectrometer (HR-ToF-AMS) (DeCarlo et al., 2006). The soluble ions of sodium and potassium
was measured by a commercial instrument (GAC-IC) equipped with an aerosol collector and detected
by ion chromatography (Dong et al., 2012). The particle number size distribution (PNSD) was
measured by a scanning mobility particle sizer (SMPS, TSI 3938). The aerosol surfaces area was
calculated based on the size distribution measurement and corrected to wet particle-state by a
hygroscopicity growth factor, with a total uncertainty of determining wet aerosol surface areas by ~30%
(Liu et al., 2013). VOCs were measured by Proton Transfer Reaction Time-of-Flight Mass





Spectrometry (PTR-MS)(Wu et al., 2020;He et al., 2022) and an automated gas chromatograph
equipped with mass spectrometry or flame ionization detectors (GC-MS). A commercial instrument
(Thermo Electron model 42i) was used to monitor $NO_x$. $O_3$ was measured by a commercial instrument
using ultraviolet (UV) absorption (Thermo Electron 49i). $PM_{2.5}$ was measured by a Tapered Element
Oscillating Microbalance (TEOM, 1400A analyzer). $SO_2$ and CO were measured by commercial
instruments (Thermo Electron 43i and 48i). In addition, the meteorological parameters were available
during the measurement. Photolysis frequencies were determined by a spectroradiometer (Bohn et al.,
2008). The aerosol liquid water content (ALWC) is calculated from the ISORROPIA-II
thermodynamic equilibrium model (Clegg et al., 1998). We used the reverse mode in ISORROPIA-II
with the input of water-soluble ions along with ambient temperature ($T$) and relative humidity (RH).
Given the high RH in this campaign, we ran the model by assuming aerosol phase were metastable.
**Table 1.** Summary of the information about observed gas and particle parameters during the campaign.

| Species | Limit of detection | Methods | Accuracy |
|---|---|---|---|
| $N_2O_5$ | 6.0 pptv ($3\sigma$, 1 min) | FIGAERO-ToF-CIMS | $\pm 30\%$ |
| $ClNO_2$ | 4.3 pptv ($3\sigma$, 1 min) | FIGAERO-ToF-CIMS | $\pm 30\%$ |
| NO | 60 pptv ($2\sigma$, 1 min) | Chemiluminescence | $\pm 20\%$ |
| $NO_2$ | 0.3 ppbv ($2\sigma$, 1 min) | Mo convert | $\pm 20\%$ |
| $O_3$ | 0.5 ppbv ($2\sigma$, 1 min) | UV photometry | $\pm 5\%$ |
| VOCs | 0.1 ppbv (5 min) | PTR-ToF-MS | $\pm 30\%$ |
| VOCs | 20-300 pptv (1 h) | GC-FID/MS | $\pm 20\%$ |
| $PM_{2.5}$ | 0.1 $\mu g\,m-3$ (1 min) | TEOM | $\pm 5\%$ |
| CO | 4 ppbv (5 min) | IR photometry | $\pm 5\%$ |
| $SO_2$ | 0.1 ppbv (1 min) | Pulsed UV fluorescence | $\pm 10\%$ |
| HCHO | 25 pptv (2 min) | Hantzsch fluorimetry | $\pm 5\%$ |
| PNSD | 14 nm -700 nm (4 min) | SMPS | $\pm 20\%$ |
| Aerosol composition | $<0.16\ \mu g\,m^{-3}$ (30 min) | GAC-IC | $\pm 30\%$ |
| $PM_1$ components | 0.15 $\mu g\,m^{-3}$ (4 min) | HR-ToF-AMS | $\pm 30\%$ |
| Photolysis frequencies | Varies with species (20 s) | Spectroradiometer | $\pm 10\%$ |


**2.3 Box model setup**
A zero-dimensional chemical box model constrained by the field campaign data was applied to
simulate the $ClNO_2$ chemistry. The box model was based on the Regional Atmospheric Chemical
Mechanism version 2 (RACM2) described in Goliff et al. (2013), and chloride chemical mechanism
were added (Wang et al., 2017b;Tan et al., 2017). Briefly, chloride chemistry was adapted to RACM2
from the modifications to Master Chemical Mechanism (Xue et al., 2015), and the oxidation products
from reactions between lumped VOC species and chloride radicals were adapted from those of OH





oxidation from RACM2. $j(ClNO_2)$ was calculated according to the NASA-JPL recommendation based
on the work by Ghosh et al. (2012). The impact of $O_3$ by $ClNO_2$ chemistry was assessed by differing
the results of two scenarios with or without the constraints of the observed $ClNO_2$ in the model
simulation. For the reaction rate constant of the lumped species with Cl, the fastest value from different
species was used to represent the upper limit of the impact of chloride chemistry. The model was
constrained by the observed $ClNO_2$, NOx, $O_3$, CO, VOCs (assignment to RACM2), photolysis
frequencies, ambient temperature and pressure. The model runs were from 29 September to 17
November, 2019 with most of the measurement data taken accounted for, and with a two-days spin-
up. The lifetime of the input trace gases corresponds to a deposition velocity of 1.2 cm s-1 with an
assumed boundary layer height of 1000 m, and the model-generated species was set to 24 hours
lifetime due to the loss caused by the dry deposition. The input data were averaged and interpolated to
1 hour of resolution.
**3. Results and discussions**
**3.1 Overview of measurement**
Figure 1 shows time series of $ClNO_2$ and relevant trace gases, particles and meteorological parameters
during the measurements. In this campaign, the meteorological condition featured high temperature
($24.7 \pm 3.8$ °C) and high humidity ($62.1\% \pm 15.6\%$), low wind speed ($1.5 \pm 0.8$ m s$^{-1}$), and the dominant
air flow were from north and northwest. Compared to those with previously measurements at the same
site in January 2017 (Yun et al., 2018b), the temperature was higher and relative humidity was lower
during the measurements. The average and maximum concentration of particulate matter ($PM_{2.5}$) was
$47.6 \pm 19.3$ μg m$^{-3}$ and 138 μg m$^{-3}$, respectively, which is significantly lower than that observed in
January 2017, with a maximum up to 400 μg m$^{-3}$. The dominant air pollutant was $O_3$ with hourly
campaign maximum and the average mean daily maximum 8-hour $O_3$ (MDA8 $O_3$) of 152.8 ppbv and
$75.2 \pm 20.9$ ppbv, respectively. There were 27 days out of 53 days with the hourly maximum of $O_3$
exceeded the Chinese national air quality standard (200 μg m$^{-3}$, equivalent to 93 ppbv), suggesting
severe ozone pollution during the measurement period in PRD region. $NO_2$ concentration was also
elevated with $21.0 \pm 10.4$ ppbv on campaign average. The concurrent high $O_3$ and $NO_2$ made large
nitrate radical production rate occurred with a daily average of $1.8 \pm 2.1$ ppbv h$^{-1}$. The campaign
maximum $NO_3$ production rate was observed up to 18.6 ppbv h$^{-1}$ in the afternoon at 11$^{th}$ November,
2019. However, high $NO_3$ production rate did not mean high concentrations of $NO_3$, $N_2O_5$ and $ClNO_2$
in the atmosphere, as the concentration affected by both their sources and sinks.

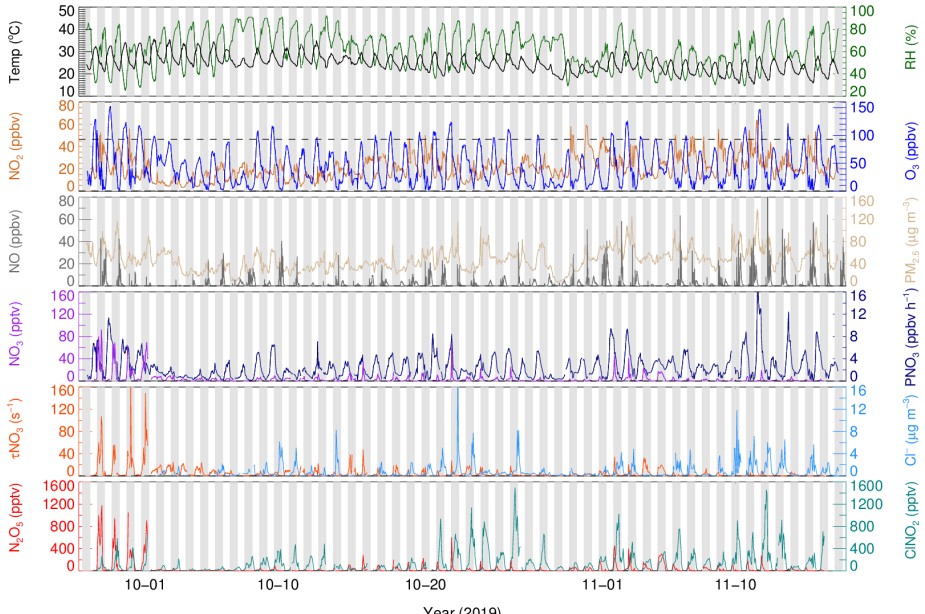


**Figure 1.** Time series of $N_2O_5$, $ClNO_2$ and relevant parameters. The grey dotted line in the $O_3$ panel denotes Chinese national air quality standard for hourly maximum $O_3$ (200 μg m$^{-3}$, equivalent to 93 ppbv). $NO_3$ radical is calculated based on a thermal equilibrium with measured $NO_2$ and $N_2O_5$.

$N_2O_5$ existed at a moderate concentration at most nights, with the daily nocturnal peaks range from <100 pptv to 1180 pptv and nocturnal average of 64 ± 145 pptv. During the nights from 27$^{th}$ – 30$^{th}$ September, 2019, the $N_2O_5$ concentration was significantly higher than other nights. The $NO_3$ lifetime, calculated by steady state method (Brown et al., 2003), was much longer in the four nights than other nights, implied relative weak sink of $NO_3$-$N_2O_5$ for the first four nights. The lifetime of $NO_3$ was < 1 minute in general (except the first four nights), indicating active $NO_3$ chemistry at this site. The $NO_3$ concentration was calculated assuming the thermal equilibrium of $NO_2$-$NO_3$-$N_2O_5$, with a possible lower bias caused by the equilibrium coefficient for reversible reactions of $NO_3$ and $N_2O_5$ ($K_{eq}$) (Chen et al., 2022). Figure 1 shows the variation of calculated $NO_3$ coincided with $N_2O_5$. Elevated $NO_3$ occurred at the first four nights with a maximum of 90 pptv (1 h time resolution), which is comparable with the reported $NO_3$ level at other sites in Pearl River Delta (Wang and Lu, 2019;Brown et al., 2016). $ClNO_2$ showed a clear diurnal variation with high level during the night. The nocturnal average and hourly maximum were 198±232 pptv and 1497 pptv, respectively. The abundance of $ClNO_2$ and $N_2O_5$ are lower than those observed at the same site in 2017, with high $N_2O_5$ and the highest value ever observed $ClNO_2$ of 3358 pptv and 8324 pptv (1-minute time resolution), respectively (Yun et al.,





2018b). High particulate chloride ion was observed in the site with $0.74 \pm 1.33 \ \mu g \ m^{-3}$ on nocturnal
average, which was higher at night with a peak in the second half of night and decrease at daytime.

**3. 2 Characterization of pollutants in different air masses.**

We noticed the air mass is highly varied during the measurements. For example, during the period of
10/02 - 10/05, the observed ozone and $ClNO_2$ were much lower than other days; while during the
period of 11/11 - 11/13, the air masses were much polluted with high $O_3$, $PM_{2.5}$ and $ClNO_2$. We
therefore plotted the backward trajectories of 24 h history of air masses arriving at the measurement
site at 500 m height at 00:00, 06:00, 12:00, 18:00 day by day. The measurement period was separated
into three patterns meteorologically according to the analysis of backward trajectories. Table 2 listed
the detailed information about the air mass classification. The air masses from northeast (and north)
was the dominant with a total of 37 days, which was characterized with the outflow of the center city
clusters of PRD and those from inland through long distance transport. We checked the pollutants of
the air masses from PRD and the north out of PRD (e.g., Hunan or Jiangxi Province), while no
significant difference was found. Therefore, we merged the two inland air masses as Type A. The
second type was from the coastal or offshore from east and southeast (Type B), which features the
outflow of coastal cities like Shenzhen and Hong Kong, which occurred on 12 days in total. The third
type was the clean air masses from the South China Sea (4 days, Type C). Figure 2 shows three cases
of each air masses mentioned above.
**Table 2.** The detailed information of three air mass types.

| Air mass type | Periods | Days |
|---|---|---|
| Type A: inland air from northeast | 09/26-10/01;10/08;10/11-10/20;10/24-11/10;11/14-15 | 37 (69.8%) |
| Type B: coast air from east | 10/06-07; 10/09-10; 10/21-23; 11/11-13; 11/16-17 | 12 (22.6%) |
| Type C: marine air from south | 10/02-05 | 4 (7.5%) |

The mean diurnal profiles of measured $NO_2$, $O_3$, $N_2O_5$, $ClNO_2$, the particle chloride content and the
ratio of chloride to sodium in the three types of air masses are shown in Figure 3, with a detailed
summary of related parameters in nocturnal medians listed in Table 3. High levels of $NO_2$ and $O_3$ were
observed in Type A and B air masses, with small difference of $NO_2$ diurnal variation during the second
half of night. In comparison, the two pollutants in Type C were much lower. If we focus on the
abundance at night, we found a large difference in $NO_2$ level with a sequence Type A > Type B > Type
C, which results in the same sequence of $NO_3$ productions in different air masses. The nocturnal $NO_2$
seems to be a good indicator of the level of pollution, that nocturnal CO, $PM_{2.5}$ and $SO_2$ also followed





this order with highest concentration in Type A. These results indicate that the most polluted air mass
came from the inland urban regions of PRD.

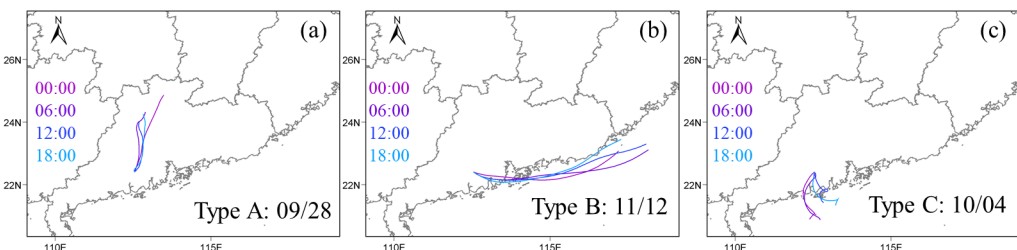


**Figure 2.** Three typical cases with air mass from different regions at 29th Sept., 12th Nov. and 4th Oct.,

respectively. Backward trajectory of 24 h history of air masses arriving at the measurement site with
500 m height at 00:00, 06:00, 12:00, 18:00.
Given the particulate chloride a precursor of $ClNO_2$, we examined its diurnal variations in the three
air mass types. The highest level of $Cl^-$ was found in Type B, and then followed by Type A and Type
C (also at night). Although the diurnal profile of $Cl^-$ in the three types is similar, the increasing rate of
$Cl^-$ during the second half of night in Type A is much slower than those in coastal and offshore air
masses. This imply a difference source of chloride, which will be further discussed in the Section 3.4.
$N_2O_5$ was observed with moderate concentration in the Type A air mass throughout the night, with a
nocturnal peak of 152.4 pptv between 20:00-21:00, while little $N_2O_5$ only occurred in the first half of
night in Type B and C with a peak of 75.9 pptv and 13.6 pptv, respectively. The concentration
difference may be attribute to two aspects. Firstly, the difference of $P(NO_3)$ results in more $N_2O_5$
produced in Type A. Secondly, compared with the air mass from coastal or offshore regions, the
nocturnal temperature and RH condition from Type A is much lower, and the loss of $N_2O_5$ may be
faster in Type B and C than that in Type A. The nocturnal median RH in Type A reached up to 67%,
while 78% and 79% in Type B and Type C, suggesting a favorable condition for heterogeneous
hydrolysis of $N_2O_5$ for all the three air mass types. The elevated $ClNO_2$ was observed in Type A and B
with a nocturnal peak of 273.6 pptv and 479.8 pptv, respectively. Significantly less $ClNO_2$ was
observed in Type C air mass with a peak of 82.6 pptv. The reason of the different levels of $ClNO_2$
observed in the three air masses types are discussed in Section. 3.4.

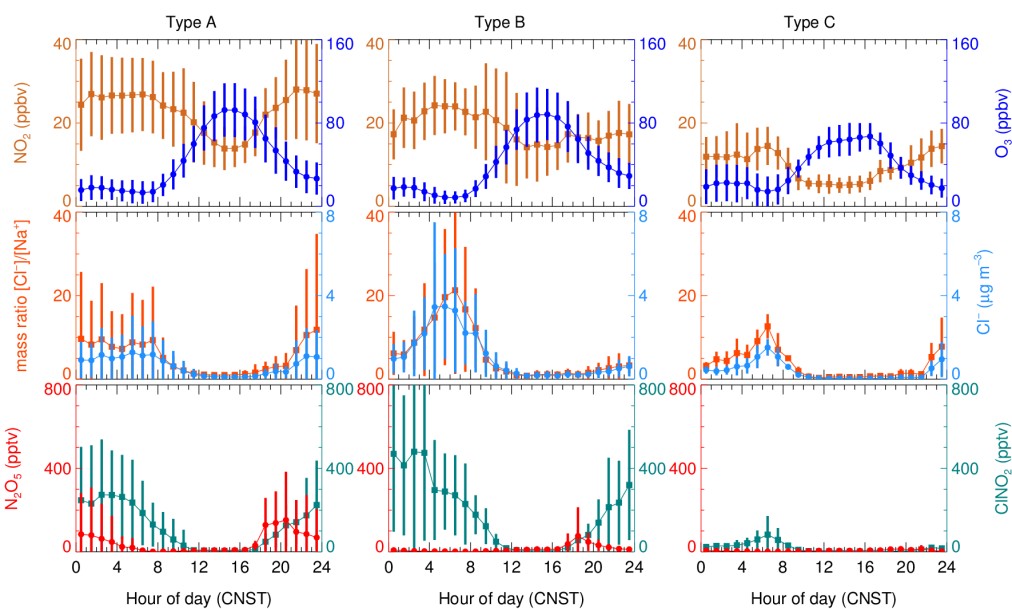


**Figure 3.** Mean diurnal profiles of $N_2O_5$, $ClNO_2$ and relevant parameters in the three types of air

masses.
**Table 3.** Statistics results (median ± standard deviation) of the related parameters in the three types
of air masses (from 18:00 to 06:00 CNST).

| Air mass | Type-A | Type-B | Type-C |
|---|---|---|---|
| RH (%) | 67.0 ± 11.9 | 78.0 ± 10.9 | 79.0 ± 9.1 |
| T (ºC) | 22.8 ± 3.0 | 23.3 ± 2.2 | 25.6 ± 1.9 |
| $ClNO_2$ (pptv) | 131.0 ± 202.8 | 162.0 ± 310.1 | 16.7 ± 21.2 |
| $N_2O_5$ (pptv) | 17.8 ± 164.9 | 6.3 ± 64.6 | 2.8 ± 9.3 |
| $Cl^-$ ($\mu g\ m^{-3}$) | 0.41 ± 1.11 | 0.56 ± 1.85 | 0.33 ± 0.51 |
| $PM_{2.5}$ ($\mu g\ m^{-3}$) | 53.0 ± 18.8 | 41.0 ± 21.8 | 32.0 ± 10.2 |
| $SO_2$ (ppbv) | 5.0 ± 4.7 | 3.4 ± 11.4 | 3.4 ± 4.7 |
| $Na^+$ ($\mu g\ m^{-3}$) | 0.12 ± 0.07 | 0.18 ± 0.09 | 0.09 ± 0.03 |
| $P(NO_3)$ (ppbv $h^{-1}$) | 1.60 ± 1.49 | 1.39 ± 1.50 | 0.69 ± 0.49 |
| $NO_2$ (ppbv) | 24.8 ± 10.9 | 18.1 ± 6.2 | 11.2 ± 5.8 |
| $O_3$ (ppbv) | 24.4 ± 21.8 | 29.5 ± 23.1 | 22.4 ± 15.2 |
| CO (ppbv) | 540.3 ± 122.3 | 448.4 ± 130.7 | 367.5 ± 89.8 |

**3.3 $N_2O_5$ uptake coefficient and $ClNO_2$ yield**

307           In line with previous studies, we estimate $N_2O_5$ uptake coefficient and $ClNO_2$ yield using the

measurements of $N_2O_5$, $ClNO_2$ and particulate nitrate (Phillips et al., 2016;Wang et al., 2018;Tham et
al., 2018). By assuming both the nocturnal enhancement of nitrate and $ClNO_2$ are mainly attributed to





$N_2O_5$ uptake processes, $ClNO_2$ yield can be solely derived by the regression analysis of $ClNO_2$ versus
particulate nitrate (Wagner et al., 2012;Riedel et al., 2013). The $\varphi ClNO_2$ can then be obtained by the
fitted regression slope (S, Eq. 1) and named as regression method.
$\varphi = 2S/(S+1)$                    (Eq. 1)

314         Combining with the data of $N_2O_5$ and aerosol surface area, the increase in $ClNO_2$ and nitrate can

be simulated simultaneously by setting the input of $N_2O_5$ uptake coefficient and $ClNO_2$ yield (named
as simulation method). The optimal $N_2O_5$ uptake coefficient and $ClNO_2$ yield are obtained
simultaneously by adjusting the two parameters until the simulation reproduces the observed increase
$ClNO_2$ and nitrate (Phillips et al., 2016;Xia et al., 2020;Tham et al., 2018). This analysis assumes only
$N_2O_5$ uptake process dominates the increase of $ClNO_2$ and nitrate, and other physicochemical
processes like vertical transportation, depositions are less important. This method requests the air mass
in the analysis duration time is relative stable and less affected by emission and transportation. In
addition, it is not valid in the case with negative changes of $ClNO_2$ and nitrate. The following selection
criteria is set to pick out the suitable plumes to meet the assumptions. Firstly, the consistent increase
trends of $ClNO_2$ and the $NO_3^-$ and clear correlation between them during the analysis duration should
be observed with a regression coefficient threshold of 0.5, which indicates the two products have the
same source. Secondary, an equivalent or faster increase of ammonium accompanied with nitrate, to
ensure insignificant degas of $HNO_3$ to the atmosphere. The observational data were averaged to 30
min for the following analysis, the time-period of each derivation ranges from 2.5 to 10 hours. Figure
4 depicts an example of the derivation on $5^{th}$ November, 2019, the stable Sa indicates stable air mass
during the analysis period. And the prediction is well reproduced the observed increase in $ClNO_2$ and
$NO_3^-$.

332         During this campaign, we carefully identified 20 plumes with clear correlations between $ClNO_2$

and particulate nitrate by the slope method ($R^2 \geq 0.5$). As shown in Table 4, the derived $ClNO_2$ yield
varied from 0.13 to 1.00 with a median of 0.45 ± 0.22 (mean value of 0.44). In the 20 plumes, we
derived $N_2O_5$ uptake coefficient and $ClNO_2$ for 12 cases in total. The results in other 8 night were not
valid due to the lack of Sa data (four nights) or producing unreasonably high results due to the observed
low $N_2O_5$ concentration near the detection limit biased the simulations. We show good consistent of
derived $ClNO_2$ yields by the two different methods. The estimated $N_2O_5$ uptake coefficient showed a
large variation and ranged from 0.0019 to 0.077 with a median of 0.0195 ± 0.0288 (mean value of
0.0317). The estimated $\gamma N_2O_5$ is within the range determined by previous field studies (Tham et al.,
2018). Specifically in China, the average level of $\gamma N_2O_5$ is comparable with those reported in urban
Beijing (Wang et al., 2017a;Wang et al., 2018), Wangdu (Tham et al., 2018), and Jinan (Wang et al.,
2017c) during the summertime, but systematically higher than those determined in China in wintertime



(Xia et al., 2021;Wang et al., 2020a;Brown et al., 2016), except the case reported on the urban canopy
of Beijing (Chen et al., 2020). McDuffie et al. (2018a) summarized the reported $\varphi ClNO_2$ based on the
observations, and we showed that the estimated average $\varphi ClNO_2$ in this study is in the middle to upper
end of the values reported globally (Xia et al., 2021;McDuffie et al., 2018a). Due to the limited data
points, we cannot distinguish the difference of $\gamma N_2O_5$ between the three air mass patterns. The $ClNO_2$
yields in Type A are slightly lower than those in Type B with an average of 0.41 and 0.47, respectively.

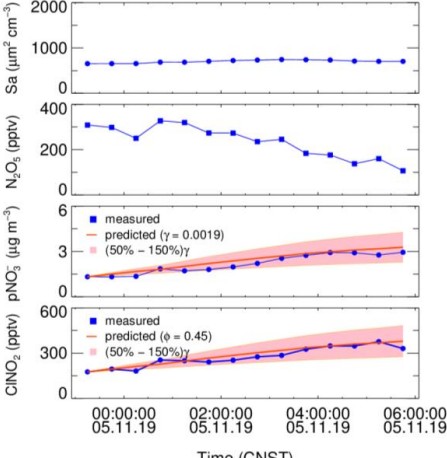


**Figure 4.** An example of the derivation of $N_2O_5$ uptake coefficient and $ClNO_2$ yield constrained by
observation of aerosol surface area, $N_2O_5$ and the enhancement of particulate nitrate and $ClNO_2$ on 5th
November, 2019. The pink region presents ±50% uncertainty of $N_2O_5$ uptake coefficient.
**Table 4.** The derived $N_2O_5$ uptake coefficient and $ClNO_2$ yields at each night.

| NO. | Period | $\gamma N_2O_5$ [a] | $\varphi ClNO_2$ [a] | $\varphi ClNO_2$ [b] | $r^2$ [b] | Type |
|---|---|---|---|---|---|---|
| 1 | 10/02 01:00-06:00 | NaN | NaN | 0.13 | 0.90 | C |
| 2 | 10/02 23:00-06:00 | NaN | NaN | 0.25 | 0.90 | C |
| 3 | 10/11 01:00-04:00 | NaN | NaN | 0.65 | 1.00 | B |
| 4 | 10/14 23:00-04:00 | 0.017 | 0.28 | 0.23 | 0.56 | A |
| 5 | 10/18 18:00-21:00 | 0.0059 | 0.42 | 0.40 | 0.90 | A |
| 6 | 10/20 20:30-23:00 | 0.045 | 0.44 | 0.47 | 0.71 | A |
| 7 | 10/21 20:30-01:00 | 0.061 | 0.52 | 0.54 | 0.90 | B |
| 8 | 10/22 22:30-05:00 | 0.066 | 0.58 | 0.61 | 0.62 | B |
| 9 | 10/24 22:00-06:00 | 0.065 | 0.26 | 0.23 | 0.74 | A |
| 10 | 10/25 21:00-02:00 | 0.077 | 1.00 | 1.00 | 0.92 | A |
| 11 | 10/28 21:00-04:00 | NaN | NaN | 0.15 | 0.74 | A |
| 12 | 11/01 21:00-23:30 | 0.022 | 0.35 | 0.32 | 0.83 | A |
| 13 | 11/02 22:00-00:30 | NaN | NaN | 0.29 | 1.00 | A |

| 14 | 11/03 18:00-06:00 | 0.0031 | 0.52 | 0.50 | 0.92 | A |
| 15 | 11/04 22:00-06:00 | 0.0019 | 0.45 | 0.47 | 0.86 | A |
| 16 | 11/08 00:00-06:00 | 0.0097 | 0.34 | 0.32 | 0.85 | A |
| 17 | 11/10 00:00-04:00 | NaN | NaN | 0.59 | 0.80 | A |
| 18 | 11/11 22:00-04:00 | NaN | NaN | 0.53 | 0.50 | B |
| 19 | 11/12 22:00-04:00 | NaN | NaN | 0.42 | 0.62 | B |
| 20 | 11/13 21:00-00:00 | 0.0070 | 0.70 | 0.75 | 0.92 | B |

Note: [a] the values of $\gamma N_2O_5$ and $\varphi ClNO_2$ are derived by simulation method; [b] the $\varphi ClNO_2$ and the
correlation coefficient ($R^2$) between $ClNO_2$ and particulate nitrate are derived by regression method,
the data was filtered with a correlation coefficient obtained from linear fitting threshold of 0.5.
To gain insight into the factors governing the $N_2O_5$ uptake and $ClNO_2$ formation processes, the
estimated $\gamma N_2O_5$ and $\varphi ClNO_2$ were compared with those predicted from complex laboratory-derived
and field-derived parameterizations. An aqueous inorganic iconic reaction mechanism once raised by
Bertram and Thornton (2009) and established a volume-limited parameterization by considering the
aerosol volume, surface area, nitrate content, ALWC, and chloride content (named BT09, Eq. 8).
$$\gamma_{BT09} = \frac{4H_{aq}Vk}{CS_a}\left(1 - \frac{1}{1 + \frac{k_3[H_2O]}{k_{2b}[NO_3^-]} + \frac{k_4[Cl^-]}{k_{2b}[NO_3^-]}}\right) \tag{8}$$
Where $H_{aq}$ is Henry's law coefficient of $N_2O_5$, $V$ is the aerosol volume; $k$ is equal to $1.15\times10^6$-(1.15
$\times10^6)^{\exp(-0.13[H2O])}$; $k_3/k_{2b}$ is the ratio of reaction rate of $H_2O$ versus $NO_3^-$ to $H_2ONO_2^+$ that was set to
0.06, and $k_4/k_{2b}$ is the ratio of reaction rate of $Cl^-$ versus $NO_3^-$ to $H_2ONO_2^+$ that was set to 29 (Bertram
and Thornton, 2009). The mean values of particulate volume to surface ratio (V/Sa) was measured. A
simple parameterization (EJ05) considered the effect of enhanced RH and temperature on $N_2O_5$ uptake
was also included (Evans and Jacob, 2005). In addition, the recently established empirical
parameterization based on the same framework (Eq. 8, named Yu20), optimized some parameters
according to the meta-analysis of five field measurements in China by Yu et al. (2020), also assessed
in the study. Figure 5(a) shows the correlation of estimated $\gamma N_2O_5$ versus the parameterization. All the
three parameterizations fail to predict the high values. The simple parameterization of EJ05 had the
best performance with a high correlation and a consistent prediction of the median value. While other
two parameterizations, BT09 and Yu20, underestimated the observed $\gamma N_2O_5$. Figure 6(a-h) show the
dependence of the observed $\gamma N_2O_5$ on the factors reported in previous literatures that possibly alert the
processes of $N_2O_5$ uptake and $ClNO_2$ formation. We show that $\gamma N_2O_5$ highly correlated with the
ambient RH as well as liquid water content, confirming the critical role of water content in $N_2O_5$ uptake
and explained the reason why EJ05 had a good performance. The dependence of $\gamma N_2O_5$ on nitrate mass



concentration does not follows the rule of nitrate suppressing effect (Wahner et al., 1998), which may
be due to the covariance of nitrate and liquid water content. With respect to other factors, insignificant
impacts on the $N_2O_5$ uptake are obtained.

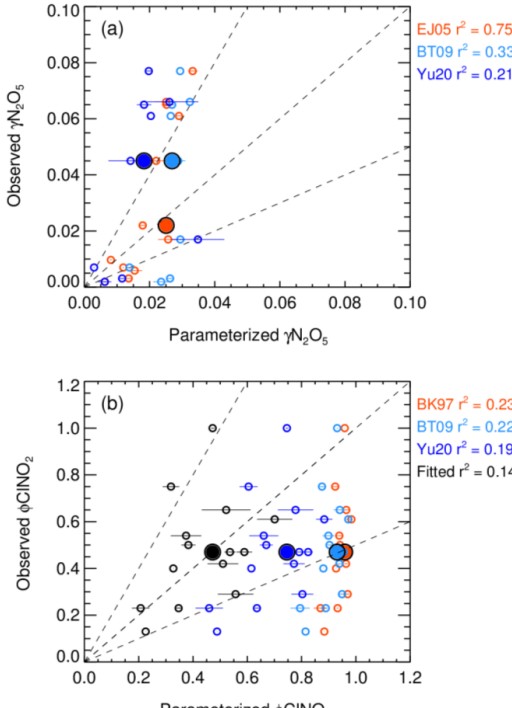


**Figure 5.** The inter-comparison of observation and parameterization of $N_2O_5$ uptake coefficient (a)
and $ClNO_2$ yield (b). The larger size of solid dots represents the median results. The parametrizations
of EJ05, BT09, Yu20 and BK97 cited from Evans and Jacob (2005), Bertram and Thornton (2009), Yu
et al. (2020), Behnke et al. (1997), respectively. The fitted $ClNO_2$ yield (colored by black) in panel (b)
shows the best fitting result in the study by adopting the $k_4/k_3$ of 32.0.
Bertram and Thornton (2009) also proposed a $ClNO_2$ yield parameterization method that
considering the ratio of ALWC and chloride content (Eq. 9), here the $k_4/k_3$ was the ratio of reaction
rate of $H_2ONO_2^+$ versus $Cl^-$ to $H_2O$ and adopted as $483 \pm 175$. Behnke et al. (1997) determined this
ratio of $836 \pm 32$, while it is estimated to be $105 \pm 37$ in Yu et al. (2020).
$$\varphi_{BT09} = \left( \frac{[H_2O]}{1 + k_4/k_3[Cl^-]} \right)^{-1} \qquad (9)$$
Figure 5(b) shows that all the predicting $ClNO_2$ yield based on the abovementioned parameterizations
overestimated the observations. The performance of the parameterization schemes of BK97 and BT09





based on the model aerosol conditions with an overestimation up to ~100% are expected and consistent
with previous studies, which may be caused by the unaccounted potentially competitive effect of other
species like organics, sulfate for the $NO_2$ intermediate (McDuffie et al., 2018a;Staudt et al., 2019;Xia
et al., 2021;Wang et al., 2017d). Although the empirical parameterization (Yu20) based on field
observations improved the prediction and narrowed the gap effectively, the overestimation is still large
with an average of ~50%, which indicated that the yield are more strongly suppressed than those
observed in the campaigns of Yu et al. (2020). The factor 32.0 ($k_4/k_3$ in Eq. 9) was derived by iterative
algorithms to achieve the best consistent between the observed and parameterized $ClNO_2$ yield, which
is smaller than the Yu20 parameters by factors of 3.3. We examined the relationships of $ClNO_2$ yields
with aerosol water content and other aerosol compositions as shown in Figure 6(i-p). We show that
φ$ClNO_2$ only weakly correlated with the content of chloride (including the mass ratio and fraction in
$PM_{2.5}$) and the molar ratio of chloride to water, confirmed the dependence found in laboratory studies.
However, we did not find the dependence of the yields with aerosol organic or sulfate, as well as the
RH and water alone in the campaign, implying the $ClNO_2$ yield mechanism is much more complicated
than the laboratory conditions.

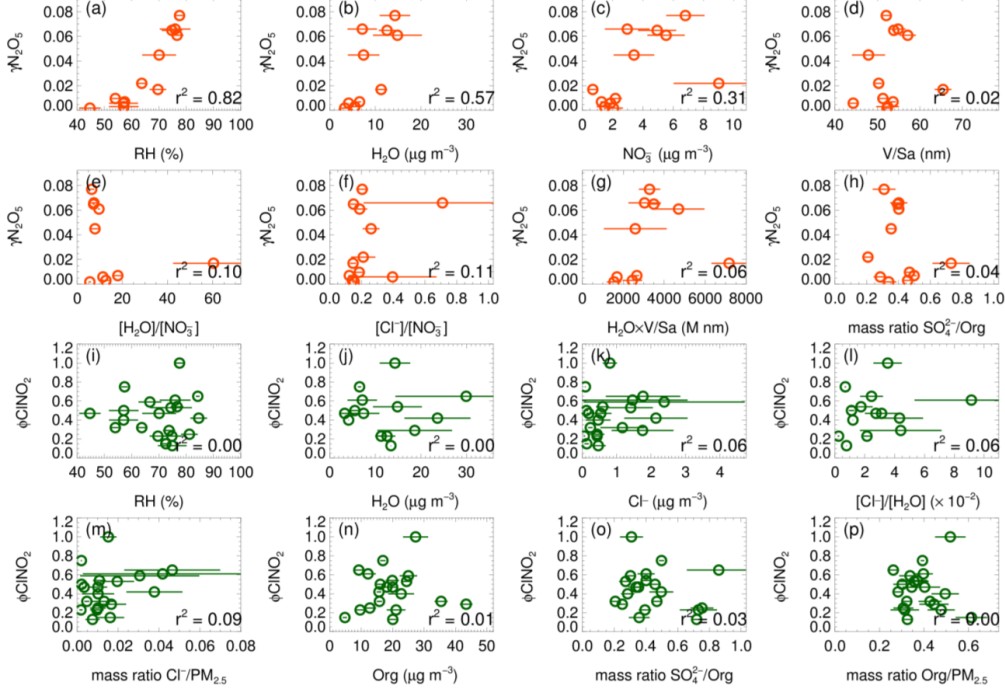

**Figure 6.** The estimated $N_2O_5$ uptake coefficient and $ClNO_2$ yield versus related parameters.

### 3.4 The factors influence ClNO$_2$ formation

The ClNO$_2$ formation can be largely affected by the budget of NO$_3$-N$_2$O$_5$ and N$_2$O$_5$ uptake processes. The variation of NO$_3$ loss by VOC and NO alert the NO$_3$ loss distribution by N$_2$O$_5$ uptake and ClNO$_2$ formation indirectly. Figure 7 shows the correlation between daily median ClNO$_2$ and mass concentration of chloride, PM$_{2.5}$ and NO$_3$ production rate for the three types of air masses. Due to the limited dataset of type C, the correlation analysis may not make sense, therefore, we did not take type C into consideration in detailed discussion. We show that the mass concentration of chloride also showed a correlation coefficient with ClNO$_2$ by 0.66 and 0.31 for type A and B, respectively. Furthermore, the mass concentration of PM$_{2.5}$ correlated reasonably with the ClNO$_2$ formation with the correlation coefficient of 0.39 and 0.62 for type A and B, respectively. But, the levels of ClNO$_2$ demonstrate little relationship with the nitrate production rate. This is quite different from the results observed in United Kingdom, where the ClNO$_2$ levels are mainly controlled by NO$_2$ and O$_3$, rather than by the N$_2$O$_5$ uptake processes (Sommariva et al., 2018).

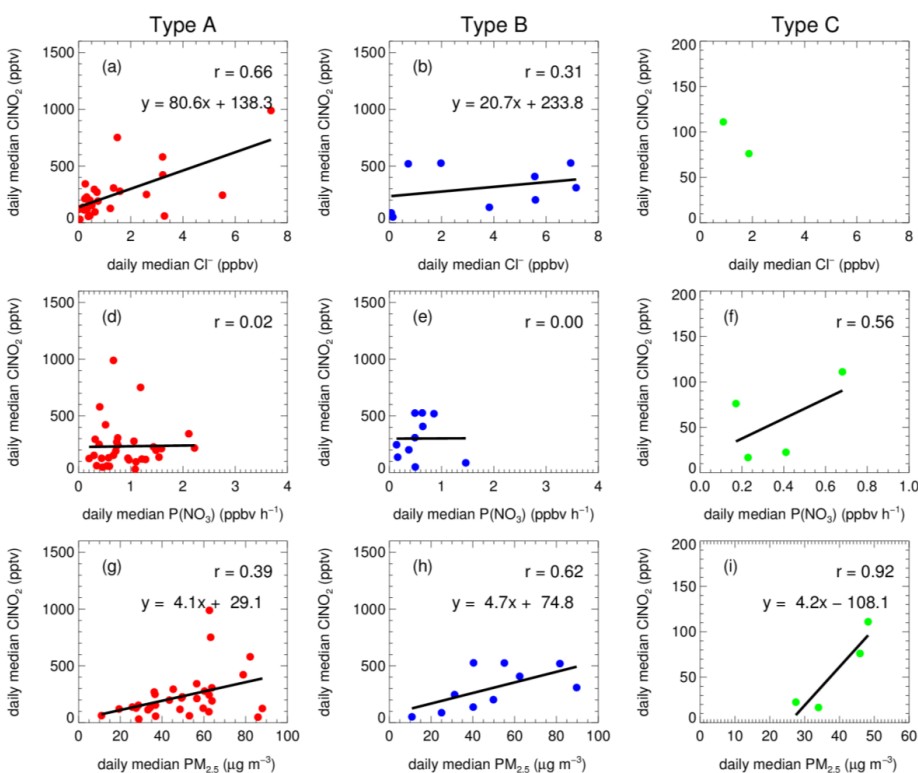

**Figure 7.** The functional dependence of daily median of ClNO$_2$ on particulate chloride, nitrate radical production rate and PM$_{2.5}$ in the air mass of Type A (a, d, g), Type B (b, e, h) and Type C (c, f, i).



The low correlation between $ClNO_2$ and $NO_3$ production rate is within expectations. In general, the
production of nitrate radical controls the total budget of $N_2O_5$, if $N_2O_5$ uptake dominated the sink of
$NO_3$, as the result the $N_2O_5$ uptake and its products would show good correlation with $NO_3$ production
rate. But in fact, $NO_3$ loss can also be affected by other loss pathways, like the reactions with NO and
VOCs. In many cases, the $NO_3$ loss is dominated by VOC or NO, that means the $ClNO_2$ formation is
suppressed. If the two loss pathways are highly varied due to irregular emissions, then the relationship
between $ClNO_2$ and $NO_3$ production rate would be less correlated. We confirmed large variations of
NO and VOC (not shown) in hourly and daily scales, which means the proportion of $N_2O_5$ uptake to
the total loss of $NO_3$ is highly varied correspondingly. Overall, the low correlation in the study
indicated that the $ClNO_2$ formation through $N_2O_5$ uptake is not limited by $NO_3$ formation processes,
at least in Type A and B. With respect to the air mass Type C, $ClNO_2$ showed correlation with $P(NO_3)$
with the correlation coefficient of 0.56.
As the precursor of $ClNO_2$, higher concentrations of particulate chloride result in high $ClNO_2$ yield
from $N_2O_5$ uptake to some extent, as evidenced by our field observation (Figure 6) and previous
laboratory studies (Bertram and Thornton, 2009;Roberts et al., 2009;Ryder et al., 2015). High $PM_{2.5}$
concentrations usually provide more aerosol surface area to promote $N_2O_5$ uptake. The close
relationship between $ClNO_2$ and $PM_{2.5}$ indicate that aerosol surface area, most likely, is a critical factor
that limited $ClNO_2$ formation. The proportion of nitrate in the total $PM_{1.0}$ was small with an average
of 10.4%, therefore the correlation of $ClNO_2$ and $PM_{2.5}$ cannot attribute to the covariance between
nitrate and $PM_{2.5}$. In addition, the $ClNO_2$ level in the air mass of Type B show higher correlation to
both $Cl^-$ and $PM_{2.5}$ than type A, suggesting that the $ClNO_2$ formation in Type B is more effectively
affected by the levels of chloride and $PM_{2.5}$.
Recently model simulation indicated that the $ClNO_2$ chemistry level is sensitive to the emission of
chloride in PRD (Li et al., 2021). In this study, a question raised that where is the source of chloride?
The mass ratio of $Cl^-/Na^+$ is often used as an indicator of sea salt or anthropogenic sources to chloride
with a threshold of 1.81 (Yang et al., 2018;Wang et al., 2016). High ratio means the particulate chloride
affected by anthropogenic emission rather than sea salt. We determine that the mean mass ratios of $Cl^-$
to $Na^+$ are 5.3, 6.3 and 3.1 in Type A, B and C, respectively (Figure 3). This indicated that $PM_{2.5}$
sampled during the campaign was not strongly influenced by fresh sea salt aerosols. In the three types,
the Type C air mass had a lowest ratio and may be influenced by both sea salt and anthropogenic
emissions, which seems reasonable since it come from South China Sea. If we assume that Type A air
mass is free of sea salt and only influenced by anthropogenic activities, the higher ratio implies more
intensive chloride source in Type B. The correlation between particulate chloride and some possible
indicators, including $K^+$, benzene, $SO_2$, CO, acetonitrile ($CH_3CN$), were examined day by day. Figure



8 shows the max correlation coefficient ($R^2$) in each day with a threshold of 0.5. We filtered out 39 out
of 46 days during this campaign with a fraction of 85%. Among the 39 days, a total of 11 days is
associated with strongest correlation between Cl⁻ and benzene, which is typically come from industrial
emissions. Cl⁻ also correlated with $K^+$, CO and $CH_3CN$ in 19 day in total, implies potential
contributions from biomass burning emissions. In total of 9 days for highest correlations of Cl- with
$SO_2$ indicated power plants emissions may also contributed to Cl⁻ emission. We summarized that the
source of chloride may be highly varied from different anthropogenic activities including biomass
burning, industrial processes as well as power plants. The statistic results in Table 5 suggest that the
Cl⁻ in air mass of Type A were affected by various sources, especially related to the sources associated
with $K^+$, benzene and $CH_3CN$; the Cl⁻ in Type B was mainly contributed by the similar source of CO,
and Type C was only affected by power plants emissions. In addition, Figure 8 showed that there are
2 days that the correlations between Cl⁻ and $Na^+$ exceeded the max of the selected anthropogenic factor
matrix, indicated that the aerosol still also impacted by sea salt to some extent.

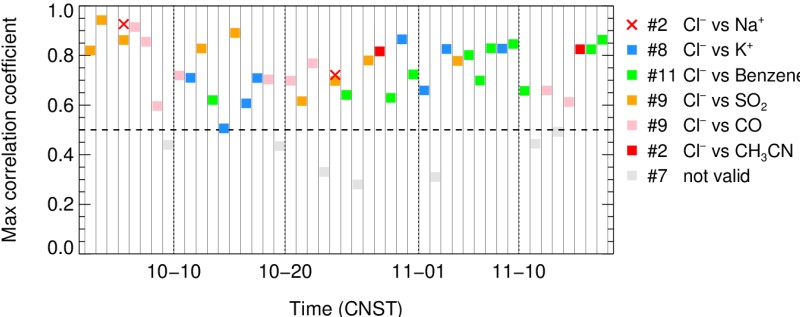

**Figure 8.** The max correlation coefficient between particulate chloride and a selected parameter matrix
(including $K^+$, benzene, $SO_2$, CO, acetonitrile ($CH_3CN$)) in each day. The labelled number in each
legend indicates the days be the maximum, the dashed line denotes the threshold of 0.5 (39 valid days
out of 46 in total). The cross means the correlation coefficient between Cl⁻ and $Na^+$ is larger than the
max.

**Table 5.** The statistic of the days for highest factors correlated with particulate chloride in different
air mass pattern.

| factors | Type A | Type B | Type C |
|---|---|---|---|
| $K^+$ | 8 | 0 | 0 |
| Benzene | 9 | 2 | 0 |
| $SO_2$ | 5 | 1 | 3 |
| CO | 4 | 5 | 0 |
| $CH_3CN$ | 2 | 0 | 0 |



**3.5 The impacts of ClNO₂ on atmospheric oxidation**

In this section, we focus on the assessment of the impact of ClNO₂ photolysis on the source of radicals and the contribution to the atmospheric oxidation. Figure 9 shows the diurnal accumulation of ROx production rate from model simulations with ClNO₂ chemistry in the three types of air mass. The total ROx production rate was higher in Type A and then followed by Type B and C, in which photolysis of HONO, HCHO, O₃ and OVOCs had large contributions. Cl radical, liberated by ClNO₂, enhanced little ROx production, with a morning peak contribution of 1.3%, 2.2% and 1.8% for Type A, B, C, respectively (08:00-09:00). The contribution of ClNO₂ photolysis to the production of ROx is less than 1% on daytime averaged, similar to the results obtained in winter Shanghai (Lou et al., 2022) as well as North China (Xia et al., 2021), and much lower compared to    previous studies reported in summer time in both north and south China (Tan et al., 2017;Wang et al., 2016;Tham et al., 2016). In addition, we noticed that the significant role of OVOCs (including photolysis and reacts with O₃) in producing ROx at this site, especially in the Type A and B air mass. This result is consistent with that constrained by observed OVOCs in Guangzhou City (Wang et al., 2022c).

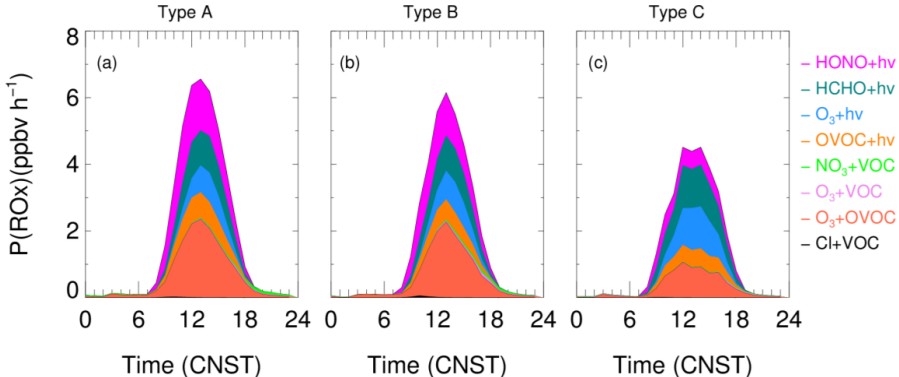

**Figure 9.** The diurnal cycle and distribution of ROx production rate in the three types air masses.

Figure 10 shows the enhancement of OH, HO₂ and RO₂ radicals with the consideration of ClNO₂ chemistry. The enhancement of the three radicals peaked in the morning. On average, OH concentration was enhanced by 1.5% to 2.6% in different air masses. The percentage of enhancement for HO₂ radical was 1.9% to 4.6%, whereas the enhancement for RO₂ was a little bit higher (3.0% to 6.8%). In general, the enhancement of radicals was more significant in Type B than other two types of air masses, which is related to elevated ClNO₂ concentrations for these air masses. Low ClNO₂ and other radical precursors led to an earlier enhancement peak (08:00-09:00) in Type C and lasted a short time period. Although the increase peak occurred later at 09:00-10:00 for the air mass of Type A and

Type B, the increase lasted for a longer time and had a longer effect. Overall, daytime OH, $HO_2$ and
$RO_2$ enhanced by 1.0%, 2.0%, and 3.0% on campaign average.

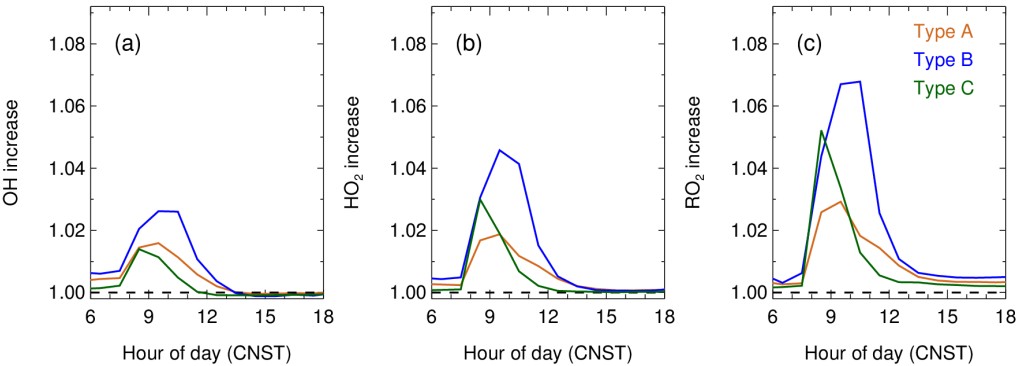


**Figure 10.** The diurnal cycle on the enhancement of OH (a), $HO_2$ (b), $RO_2$ (c) by $ClNO_2$ chemistry in
the three air mass patterns.

513       Figure 11 depicts the integral enhancement of $O_3$ production by $ClNO_2$ photolysis varied from less

than 0.1 ppb to 4 ppb day by day, with a percentage of <1% to 4.9% with a median of 0.8%. Our results
are comparable with the winter case in North China (Xia et al., 2021). The next day $O_3$ enhancement
was highly correlated with the level of $ClNO_2$ with the correlation coefficient of 0.7. The net $O_3$
production enhanced by 0.70 ppbv $h^{-1}$ (0.9%), 1.02 ppbv $h^{-1}$ (1.9%), 0.24 ppbv $h^{-1}$ (0.6%) on daytime
accumulation in Type A, B, C, respectively, which is consistent with the nocturnal level of $ClNO_2$ in
the three air masses presented in Table 3.

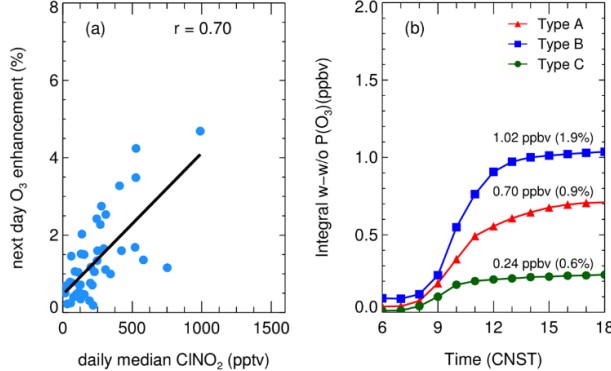


**Figure 11.** (a) The correlation of daily median $ClNO_2$ (18:00-06:00) and its impact on next day net $O_3$
production enhancement during the campaign; (b) the average contribution of daytime integral $O_3$ by
$ClNO_2$ mechanism in the three types of air masses.



Previous studies suggest that chlorine radicals from $ClNO_2$ photolysis may contribute significantly
to the oxidation of some VOCs species, especially for long-chain alkanes(Shi et al., 2020;Wang et al.,
2022b). The oxidation of long-chain alkanes (C10~14 n-alkanes) by chloride and OH radicals during
the morning hour (08:00 - 09:00) were also evaluated based on modeled oxidants concentration. We
observe small contributions of chloride radical with a percentage of 3.2%, 3.7% and 4.2% for n-decane,
n-dodecane and n-tetradecane, respectively. The contributions reduced to <1% on daytime average.
We also checked the role of chloride in short-chain alkanes oxidation, obtaining an even smaller
contribution than the long-chain alkanes. Therefore, we concluded that, chloride radicals liberated by
$ClNO_2$ photolysis, is not critical to the oxidation of alkanes compared with OH oxidation during the
campaign. We note that several studies reported other sources produced large amount of chloride
radicals like $Cl_2$ (Liu et al., 2017;Xia et al., 2020), BrCl (Peng et al., 2021), the daytime reaction of
HCl with OH (Riedel et al., 2012;Eger et al., 2019;Li et al., 2019). However, it is not possible to assess
the overall impacts by constraining all precursors of chloride radical, which may warrant further
investigation by more comprehensive field studies equipped with the instruments for detecting these
species.
**4. Conclusion**
An intensive field study in Pearl River Delta took place during a photochemical pollution season from
Sept. 26th to Nov. 17th, 2019, providing a comprehensive observation dataset to understand the $ClNO_2$
chemistry and its impacts on the air quality. We found that the air masses highly varied from different
regions and divided in three types according to the results of backward trajectory. Two of them, air
mass from northern and northeastern inland cities and the eastern coastal regions, features polluted
with elevated $O_3$ and related trace gases like NOx and CO. Correlation analysis showed that $ClNO_2$
formation is limited by chloride availability and $PM_{2.5}$ concentrations (mostly due to aerosol surface
area) at this site. In general, we observed a wide variation for determining factors of $ClNO_2$ formation
in different kinds of air masses.
We estimated the $N_2O_5$ uptake coefficients and $ClNO_2$ yield during this campaign and assessed the
performance of previous parameterizations schemes. The newly developed observation-based
empirical parameterization was also checked and showed an overall underestimation. We showed the
$\gamma N_2O_5$ only strongly correlated with RH, and the parameterization proposed by Evans and Jacob (2005)
showed a considerable consistent with the observation. The $ClNO_2$ yield only showed weak correlation
with the content of particle chloride, and the exist parameterizations systematically overestimated the
yield.
The particulate chloride mainly originated from anthropogenic emissions rather than sea salt.



However, the specific contributing source of chloride in this region cannot be determined, due to the
varying correlation relationship with different kinds of anthropogenic emission indicators day by day.
We can only infer that the air mass of Type A affected by most complicated anthropogenic emissions
including biomass burning, power plants as well as the even possible usage of industrial solvents. This
result highlights the $ClNO_2$ chemistry may be triggered by many kinds of anthropogenic activities in
the PRD regions (Wang et al., 2016;Yang et al., 2018). The sources of particulate chloride warrant
further detailed exploration using the dataset along with other observations in this region.

564        In the end, we investigate the impacts of $ClNO_2$ chemistry on atmospheric oxidation by a box model.

It is demonstrated that chloride radicals liberated by $ClNO_2$ chemistry had a relatively small
contribution to the following daytime level of OH, $HO_2$, and $RO_2$ radicals, as well as a small
enhancement of $O_3$ and ROx production in all the three types of air masses. The impacts of $ClNO_2$
chemistry were larger in the Type B than that of Type A. Overall, the small contribution of $ClNO_2$
chemistry in PRD region may be due to the limited $ClNO_2$ produced by $N_2O_5$ uptake processes, and
other strong primary sources of radicals weakened its contribution indirectly. Given complex source
of particulate chloride, we call for more field investigations to address the chloride chemistry and its
roles in air pollutions in China.
**Data availability.** The datasets used in this study are available from the corresponding author upon
request (byuan@jnu.edu.cn).
**Author contributions.** H.C.W. and B.Y. designed the study. E.Z, X.X.Z. and H.C.W. operated and
calibrated the CIMS, H.C.W. analyzed the data, H.C.W. and B.Y. wrote the manuscript with inputs
from all coauthors.
**Competing interests.** The authors declare that they have no conflicts of interest.
*Acknowledgements*. This work was supported by the National Natural Science Foundation of China
(grant No. 41877302, 42175111, 42121004), Guangdong Natural Science Funds for Distinguished
Young Scholar (grant No. 2018B030306037), Key-Area Research and Development Program of
Guangdong Province (grant No. 2019B110206001), and Guangdong Innovative and Entrepreneurial
Research Team Program (grant No. 2016ZT06N263). This work was also supported by Special Fund
Project for Science and Technology Innovation Strategy of Guangdong Province (Grant
No.2019B121205004). The authors gratefully acknowledge the Jinan University science team for their





technical support and discussions during this campaign. We thank for the NOAA Air Resources
Laboratory for providing the HYSPLIT model.
**Appendix**
**A1. The calibration of CIMS**
The calibration of $ClNO_2$ measurement sensitivity has been introduced in Wang et al. (2022a). In brief,
a nitrogen flow (6 mL min$^{-1}$) containing 10 ppmv $Cl_2$ was passed over a slurry containing $NaNO_2$ and
NaCl to produce $ClNO_2$ (Thaler et al., 2011), and NaCl was included in the slurry in order to minimize
the formation of $NO_2$ as a byproduct. The mixed flow containing $ClNO_2$ was then conditioned to a
given RH and sampled into the CIMS instrument. To quantify $ClNO_2$, the mixed flow was delivered
directly into a cavity attenuated phase shift spectroscopy instrument (CAPS, Model N500, Teledyne
API) to measure background $NO_2$ concentrations or through a thermal dissociation tube at 365 °C to
fully decompose $ClNO_2$ to $NO_2$, and the total $NO_2$ concentrations were then determined using CAPS.
The differences in the measured $NO_2$ concentrations with and without thermal dissociation was
equivalent to $ClNO_2$ concentrations. The CAPS instrument had a detection limit of 0.2 ppbv in 1 min
for $NO_2$ and an uncertainty of ~10%. To calibrate CIMS measurements of $N_2O_5$, a humidity adjustable
mixed flow containing stable $N_2O_5$, which was produced via $O_3$ oxidation of $NO_2$, was sampled into
the CIMS instrument to obtain a normalized humidity dependence curve of $N_2O_5$. While the
concentration of $N_2O_5$ source is not quantified due to the absence of a $N_2O_5$ detector, so we delivered
the $N_2O_5$ source flow through a supersaturated sodium chloride solution to convert $N_2O_5$ to $ClNO_2$
with a unit efficiency at 50% RH, which is a widely used method for the calibration of $ClNO_2$ by CIMS
technique. The absolute $N_2O_5$ sensitivity at RH 50% can be realized and then scaled to other humidity
condition by the normalized $N_2O_5$ sensitivity curve determined before. The sensitivity curves for $N_2O_5$
and $ClNO_2$ to water content were shown in Figure A1. Figure A2 shows the high-resolution peak fitting
results of typical mass spectra at m/z 235 and m/z 208 for $N_2O_5$ and $ClNO_2$ in three air mass patterns,
respectively. The peaks of $N_2O_5$ and $ClNO_2$ were clearly resolved in the mass spectra. The peak of
$IN_2O_5^-$ can be well retrieved by separating a large adjacent peak of $C_2H_4IO_3S^-$ in the air masses affected
by marine emissions (Type B and C), which might be hydroperoxymethyl thioformate (HPMTF) from
dimethyl sulfide oxidation (Veres et al., 2020). The interference signals including $H_3INO_2S^-$ for $ClNO_2$
measurements can also be well separated in all the three air mass patterns. These results underline the
necessity and feasibility in the application of ToF analyzer in detecting $N_2O_5$ and $ClNO_2$ with iodide
CIMS.

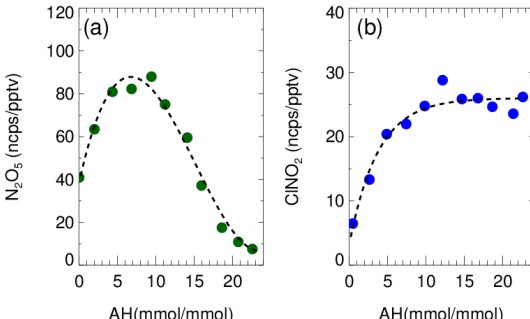


**Figure A1.** CIMS sensitivities as a function of water concentration for (a) $N_2O_5$ and (b) $ClNO_2$.

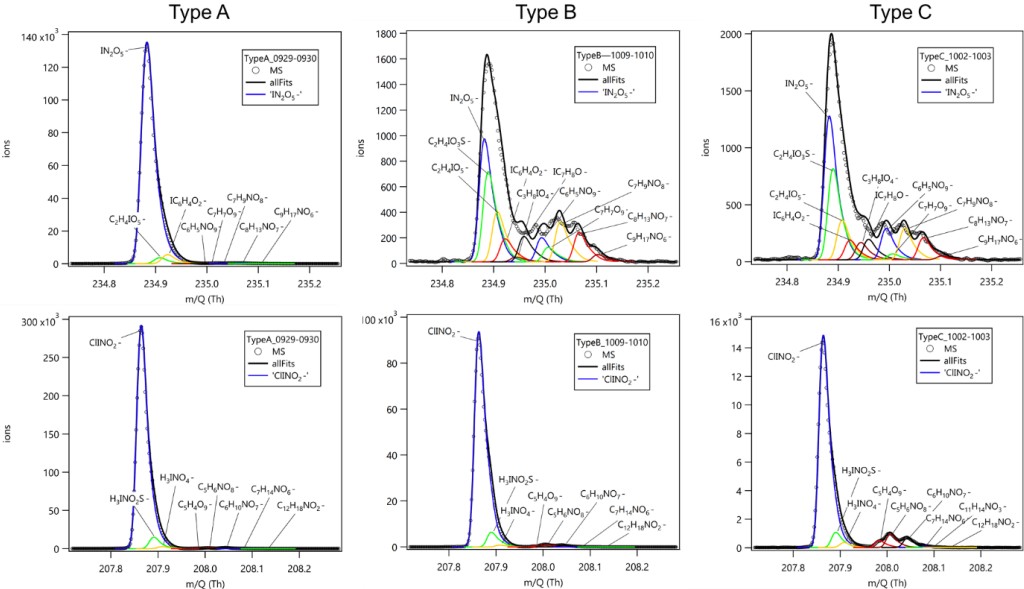


**Figure A2.** Cases of high-resolution spectra fitting for $N_2O_5$ and $ClNO_2$ by ToF-CIMS under three air
mass patterns.

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
