# Peer review of "Formation and impacts of nitryl chloride in Pearl River Delta"

_Atmospheric Chemistry and Physics, 2022_

## Referee Comment (RC1)

Comment on "Formation and impacts of nitryl chloride in Pearl River Delta" by Wang et al

Wang et al., presented field measurements of N2O5, ClNO2 and related species at a rural site of South China. Comprehensive analysis is performed regarding the sources of particulate chloride and kinetic parameters like uptake coefficient and production yield. The authors also performed model simulation to evaluate the impacts of ClNO2 on atmospheric oxidation capacity, although the impacts look minor during this observation period. The contents look rich and well fits into the scope of ACP. The methods are suitable and described with proper details. The results and conclusion are reasonable. However, the authors are suggested to show the highlight of this paper more clearly. Besides, the conclusion part can be more concise. The language polish is also necessary. Other issues are listed below. Overall, I would suggest a major revision.

Major comments:

- 1. Many of the results presented here are within expectations or can be found in previous work. The authors are suggested to show the unique value of this work. One possible way is that the authors may make more detailed comparisons with previous field observations in China and abroad and demonstrate how the results exhibited here are different from previous studies and briefly discuss why.
- 2. The authors are suggested to distinguish the concepts of "chlorine" and "chloride". In many places of this paper, "chlorine" should be used instead of "chloride". Examples include but are not limited to lines 38 and 39.
- 3. Please carefully check the language. Below are two examples: 1. Line 113, change "ubiquity" to "ubiquitous"; 2. Line 122: change "both they" to "they both".
- 4. Some doubts about the CIMS measurement. (1) Was any background measurement performed during the campaign? If yes, please present the result. If no, please clarify this point and remind readers about the uncertainty caused. (2) How often was the calibration for N2O5 and CINO2? Regular checking of the sensitivity of N2O5 or CINO2 is critical to ensure the reliable quantification of CIMS measurement. In case regular calibration is challenging in the field, the authors may investigate the record of instrument voltages, pressure, and other parameters to show how stable was the instrument during the campaign. (3) How much is the inlet artefact? As we know, N2O5 may deposit on the inlet wall and produce CINO2.

Minor comments:

- 1. Line 227-231: it is more relevant to discuss nocturnal PNO3, as N2O5 and ClNO2 are mostly produced during the night.
- 2. Line 398: change "NO2" to "NO2+".

- 3. Line 437-438: In my opinion, when ClNO2 shows no correlation with PNO3, no clear conclusion can be drawn. For example, highly variable ClNO2 yield may also cause poor ClNO2-PNO3 correlation.
- 4. Line 498: In figure 9, it shows that O3+OVOC pathway has the largest contribution to P(ROx). This is somehow surprising to me. Please double check this result.
- 5. Line 528: Again, it is surprising to me that Cl radical only have 3% to 4% contribution to long-chain alkane oxidation during morning hours. Please double check this result.
- 6. Line 533-538: Did any signals of Cl2, BrCl, Br2 detected during the campaign? Recent studies found that these more reactive halogens may have larger impacts than ClNO2.
- 7. As mentioned above, the conclusion part can be more concise.

---

## Author Comment (AC1)

Response to Editors and Reviewers

We appreciate the reviewers for their careful reading and constructive comments on our manuscript. As detailed below, the reviewer's comments are shown in black, our response to the comments is in blue. New or modified text is in red.

All the line numbers refer to Manuscript ID: acp-2022-408.

**Referee 1**

Wang et al., presented field measurements of $N_2O_5$, $ClNO_2$ and related species at a rural site of South China. Comprehensive analysis is performed regarding the sources of particulate chloride and kinetic parameters like uptake coefficient and production yield. The authors also performed model simulation to evaluate the impacts of $ClNO_2$ on atmospheric oxidation capacity, although the impacts look minor during this observation period. The contents look rich and well fits into the scope of ACP. The methods are suitable and described with proper details. The results and conclusion are reasonable. However, the authors are suggested to show the highlight of this paper more clearly. Besides, the conclusion part can be more concise. The language polish is also necessary. Other issues are listed below. Overall, I would suggest a major revision.

Thanks for the review's constructive comments. We clarified the key highlights and shortened the conclusion accordingly. More details can be found in the following response in detail.

Major comments:

1. Many of the results presented here are within expectations or can be found in previous work. The authors are suggested to show the unique value of this work. One possible way is that the authors may make more detailed comparisons with previous field observations in China and abroad and demonstrate how the results exhibited here are different from previous studies and briefly discuss why.

   We appreciate for the reviewer's suggestion to improve the value of our work. In the revised manuscript, we summarized the environmental impacts of $ClNO_2$ on the enhancement of ozone and radical formation to highlight the overall minor contribution of $ClNO_2$ chemistry to ozone formation and radical production on average conditions at different regions. Although some significant contributions from $ClNO_2$ chemistry were also observed in previous studies case by case.

   In brief, we added a paragraph and a table to discuss the impacts and the reasons as follows.

   Line 553. Table 6 summarized the observation-constrained box model simulation results about the impacts of $ClNO_2$ chemistry. The average $ClNO_2$ concentration in the observation is moderate compared with previous observations, other radical precursors (e.g., HCHO) also elevated at the same time. This leads to a large total radical and ozone production rate and a relatively minor contribution by $ClNO_2$ chemistry. This indicates that the contribution of $ClNO_2$ chemistry is indirectly affected by the budget of other radical precursors. In addition, a significant contribution by $ClNO_2$ chemistry to photochemical pollution was frequently observed in different campaigns (Tham et al., 2016; Wang et al., 2016), in which the receptor site may have aging plumes with higher $ClNO_2$ and thus larger contributions (Wang et al., 2016), suggests the large variability of $ClNO_2$ and its environmental impacts at various air masses. Here, our observations should be representative of the local condition and reflect the chemistry and impacts of $ClNO_2$ on air

pollution in the PRD region.

Table 6. The summary of the impacts of $ClNO_2$ on the next-day enhancement of ozone and radical production is based on a box model that is constrained by field observations in previous literature.

| Location | Duration | $ClNO_2$ Maximum (ppb) | Daytime average Enhancement $P(O_3)$ | Daytime average Enhancement $P(RO_x)$ | References |
|---|---|---|---|---|---|
| Heshan, China | 2019. 11 | 1.5 | 1.0%-4.9% | <2.2% | this work |
| Shanghai, China | 2020.10-11 | 0.4 | - | <1.0% | Lou et al., 2022 |
| Wangdu, China | 2014.6-7 | 2.1 | 3% | <10% | Tham et al., 2016 |
| Seoul, Korea | 2016.5-6 | 2.5 | 1.0-2.0% | - | Jeong et al., 2019 |
| Hongkong, China | 2013.11-12 | 4.7 | 11.0-41.0%[a] | - | Wang et al., 2016 |
| California, US | 2010.5-6 | 1.5 | 15.0%[b] | 17% | Riedel et al., 2014 |
| Wangdu/Beijing/ Mt. Tai, China | 2017-2018 | 1.7 | 1.3%-6.2% | 1.3%-3.8% | Xia et al., 2021 |

[a] used a box model to estimate the following evolution after the plume passing measurement site and the impacts on the next-day air quality; [b] not constrained the observed $ClNO_2$ concentration but simulated the observed maximum $ClNO_2$ case to predict the corresponding upper contribution.

2. The authors are suggested to distinguish the concepts of "chlorine" and "chloride". In many places of this paper, "chlorine" should be used instead of "chloride". Examples include but are not limited to lines 38 and 39.
   Thanks for the correction and we revised it accordingly.

3. Please carefully check the language. Below are two examples: 1. Line 113, change "ubiquity" to "ubiquitous"; 2. Line 122: change "both they" to "they both".
   We polished the language throughout the manuscript accordingly.

4. Some doubts about the CIMS measurement. (1) Was any background measurement performed during the campaign? If yes, please present the result. If no, please clarify this point and remind readers about the uncertainty caused.
   Yes, backgrounds were detected for $N_2O_5$ and $ClNO_2$ during the campaign and the results are shown in Figure. S1, for example. The green squares and line represent the average values of background and background interpolation, respectively. Which confirmed the low background of CIMS in measuring $N_2O_5$ and $ClNO_2$ in the ambient condition. In the revised manuscript, we added a brief description of the background measurement. Line 618.
   **A1. The measurement background and calibration of CIMS**
   The background measurement of $ClNO_2$ and $N_2O_5$ was performed during the campaign. Figure 1A showed an example of the background check at the beginning of the campaign, which confirmed the negligible background signal in the measurement of $ClNO_2$ and $N_2O_5$ in the ambient condition.

[Figure]

**Figure S1.** Background deduction for $N_2O_5$ (a) and $ClNO_2$ (b) (take October 20, 20:28-20:52 pm as an example)

(2) How often was the calibration for $N_2O_5$ and $ClNO_2$? Regular checking of the sensitivity of $N_2O_5$ or $ClNO_2$ is critical to ensure the reliable quantification of CIMS measurement. In case regular calibration is challenging in the field, the authors may investigate the record of instrument voltages, pressure, and other parameters to show how stable was the instrument during the campaign.

We appreciate the reviewer for the valuable suggestions. We do not often do $N_2O_5$ or $ClNO_2$ calibration during the campaign. Indeed, regular checking of the sensitivity is critical and it is also challenging to calibrate in the field. After inspection, the main parameters (pressure: voltages, etc.) of the CIMS were relatively stable, indicating that the CIMS is operating stably during the campaign. In addition, the comparison with the state of the instrument when $N_2O_5$ and $ClNO_2$ were calibrated in the laboratory showed no significant difference. Therefore, the concentration data of $N_2O_5$ and $ClNO_2$ are reliable.

Line 642. In this study, the sensitivity of the instrument was calibrated after the campaign. The main parameters (pressure: voltages, etc.) of the CIMS were checked every day and were relatively stable, indicating that the CIMS is operating stably during the campaign.

(3) How much is the inlet artifact? As we know, $N_2O_5$ may deposit on the inlet wall and produce $ClNO_2$.

We agree that $N_2O_5$ deposition on the inlet wall may produce $ClNO_2$ in the field measurement. In this campaign, a total flow rate of 8 slpm in sample mode ensured that residence time in the inlet was minimal to reduce the conversion of $N_2O_5$ to $ClNO_2$. Furthermore, throughout the campaign, there were periods when $N_2O_5$ is high up to 800 pptv while $ClNO_2$ is below 30 pptv (Figure S2), which indicates that the conversion of $N_2O_5$ to $ClNO_2$ on the inlet walls is controllable when considering the overall measurement uncertainty (Bannan et al., 2015).

Ref:

*Bannan, T. J., et al. (2015), The first UK measurements of nitryl chloride using a chemical ionization mass spectrometer in central London in the summer of 2012, and an investigation of the role of Cl atom oxidation, J. Geophys. Res. Atmos.,120, 5638–5657, doi:10.1002/2014JD022629*

[Figure]

**Figure S2.** The $N_2O_5$ level when $ClNO_2$ is lower than 30 ppt.

Minor comments:

1. Line 227-231: it is more relevant to discuss nocturnal $PNO_3$, as $N_2O_5$ and $ClNO_2$ are mostly produced during the night.

   Here we did not delete the statement about the daily average level of $PNO_3$ since we believe this parameter is valuable for intercomparison for the following related research. And we added the discussion about the nocturnal $PNO_3$ as follows.

   Line 236. At night, the nitrate radical production rate was $1.8 \pm 1.5$ ppbv $h^{-1}$ on campaign average (median, 1.4 ppbv $h^{-1}$).

2. Line 398: change "NO2" to "NO2+".

   Corrected accordingly.

3. Line 437-438: In my opinion, when $ClNO_2$ shows no correlation with $PNO_3$, no clear conclusion can be drawn. For example, highly variable $ClNO_2$ yield may also cause poor $ClNO_2$-$PNO_3$ correlation.

   Yes, this is attributed to $ClNO_2$ formation affected by many reaction steps including $NO_3$ reacts with other reactants as well as $N_2O_5$ uptake and $ClNO_2$ yield. We admit the conclusion that $ClNO_2$ formation is not limited by $NO_3$ production is a little bit arbitrary. Therefore, we changed the statement as follows. We also deleted the statement in the abstract correspondingly.

   Line 451. In addition, the variation of $N_2O_5$ uptake coefficient and $ClNO_2$ yield also results in the weak correlation between $NO_3$ production rate and $ClNO_2$ concentration. The weak correlation reflects the highly variable chemical processes from $NO_3$ production to $ClNO_2$ production in this region.

4. Line 498: In figure 9, it shows that $O_3$+OVOC pathway has the largest contribution to P(ROx). This is somehow surprising to me. Please double check this result.

   This is a very interesting question. We noticed it mainly attributed to the reaction of $O_3$ + DCB (unsaturated dicarbony) and $O_3$ + EPX (epoxide). The DCB and EPX were mainly produced from XYO and XYP (dominated by m_p_Xylene and o_Xylene), these two species were 3.55 and 1.25 ppbv on average and much higher than

previous observations in Heshan in 2014 (1.62 and 0.58 ppbv, Yang et al., ACP, 2017) and Wangdu, 2014 (0.15 and 0.11 ppbv, Tan et al., ACP, 2017). This result indicates that the role of these aromatic hydrocarbons had a large ozone potential by producing considerable OVOCs.

And we checked our model setting and input and confirmed no mistake or error in the model.

*Ref:*

*Yang, Y. D., Shao, M., Kessel, S., Li, Y., Lu, K. D., Lu, S. H., Williams, J., Zhang, Y. H., Zeng, L. M., Noelscher, A. C., Wu, Y. S., Wang, X. M., and Zheng, J. Y.: How the OH reactivity affects the ozone production efficiency: case studies in Beijing and Heshan, China, Atmos. Chem. Phys., 17, 7127-7142, 10.5194/acp-17-7127-2017, 2017.*

*Tan, Z., Fuchs, H., Lu, K., Hofzumahaus, A., Bohn, B., Broch, S., Dong, H., Gomm, S., Haeseler, R., He, L., Holland, F., Li, X., Liu, Y., Lu, S., Rohrer, F., Shao, M., Wang, B., Wang, M., Wu, Y., Zeng, L., Zhang, Y., Wahner, A., and Zhang, Y.: Radical chemistry at a rural site (Wangdu) in the North China Plain: observation and model calculations of OH, HO2 and RO2 radicals, Atmos. Chem. Phys., 17, 663-690, 10.5194/acp-17-663-2017, 2017.*

5.  Line 528: Again, it is surprising to me that Cl radical only have 3% to 4% contribution to long-chain alkane oxidation during morning hours. Please double check this result.

    Thanks for the reviewer's kind reminder, the time series of modeled Cl radical shows that the mean diurnal peak is $2.1*10^3$ molecules cm$^{-3}$. And the modeled OH radical had the mean diurnal peak of $3.9*10^6$ molecules cm$^{-3}$, which is at the moderate level of the reported observations. The above two aspects resulted in the small contribution of Cl radical to the long-chain alkane oxidation contributions (we double-checked the reaction rate constant and the calculation code). We also compared the modeled Cl and OH with the result reported in Weybourne, the UK by Bannan et al., JGR, (2017), they modeled a case with the mean diurnal peak of Cl and OH of $1.6*10^3$ molecules cm$^{-3}$ and $5.6*10^6$ molecules cm$^{-3}$, and the average fraction of alkane oxidation contributed by Cl is <1%, which also consistent with that reported in London (Bannan et al., JGR, 2015). This intercomparison provides confirmation of the plausibility of our results.

    We noticed that the C12-C14 data measured by PTR-MS was only valid from 16th Oct. to 17th Nov. 2019, therefore we updated the calculation and revised the manuscript as follows.

    Line 563. We observe small contributions of chlorine radical with a percentage of 4.3%, 4.3% and 3.8% for n-decane, n-dodecane, and n-tetradecane, respectively, during the period (Oct. 16th to Nov. 17th, 2019) when the long-chain alkanes measurement was valid. We also checked the role of chlorine radicals in short-chain alkanes oxidation, obtaining a slightly larger contribution than the long-chain alkanes, which is attributed to a relatively larger reaction rate constants between Cl with OH with respect to the short-chain alkanes. The daytime average contributions of Cl ranged from 1.4% - 1.6% varied by the chain length of the alkanes. Therefore, we concluded that chlorine radicals liberated by ClNO2 photolysis play a role in the oxidation of alkanes in the morning time, but are not critical compared with OH oxidation on the daytime average. We note that several studies reported other sources produced a large number of halogen radicals like Cl2 (Liu et al., 2017; Xia et al., 2020), BrCl (Peng et al., 2021), the daytime reaction of HCl with OH (Riedel et al., 2012; Eger et al., 2019; Li et al., 2019). These may cause more alkanes oxidized by halogen radicals. However, it is not possible to assess the overall impacts by constraining all precursors of chlorine radical in this work, which may warrant further investigation by more

comprehensive field studies equipped with the instruments for detecting these species.

*Ref.*

*Bannan, T. J., Booth, A. M., Bacak, A., Muller, J. B. A., Leather, K. E., Le Breton, M., Jones, B., Young, D., Coe, H., Allan, J., Visser, S., Slowik, J. G., Furger, M., Prevot, A. S. H., Lee, J., Dunmore, R. E., Hopkins, J. R., Hamilton, J. F., Lewis, A. C., Whalley, L. K., Sharp, T., Stone, D., Heard, D. E., Fleming, Z. L., Leigh, R., Shallcross, D. E., and Percival, C. J.: The first UK measurements of nitryl chloride using a chemical ionization mass spectrometer in central London in the summer of 2012, and an investigation of the role of Cl atom oxidation, Journal of Geophysical Research-Atmospheres, 120, 5638-5657, 10.1002/2014jd022629, 2015.*

*Bannan, T. J., Bacak, A., Le Breton, M., Flynn, M., Ouyang, B., McLeod, M., Jones, R., Malkin, T. L., Whalley, L. K., Heard, D. E., Bandy, B., Khan, M. A. H., Shallcross, D. E., and Percival, C. J.: Ground and Airborne UK Measurements of Nitryl Chloride: An Investigation of the Role of Cl Atom Oxidation at Weybourne Atmospheric Observatory, Journal of Geophysical Research-Atmospheres, 122, 11154-11165, 10.1002/2017jd026624, 2017.*

6. Line 533-538: Did any signals of $Cl_2$, $BrCl$, $Br_2$ detected during the campaign? Recent studies found that these more reactive halogens may have larger impacts than $ClNO_2$.

This is a very interesting topic and we did not notice these species before. We have not calibrated m/z 196.8 ($Cl_2$), m/z 240.8 ($BrCl$) and m/z 284.7 ($Br_2$) in the laboratory and field. In this study, we used a semi-quantitative method to infer the sensitivity of these species, which is divided into two steps: 1) obtain the dissociation voltage of the I-cluster and establish a functional relationship between the dissociation voltage and the relative sensitivity (Lopez-Hilfiker et al., 2016); 2) establish the relative transmission efficiency curve of the CIMS through laboratory experiments in order to correct the mass discrimination effect (Heinritzi et al., 2016). In this way, we get the concentrations of $Cl_2$, $BrCl$ and $Br_2$ as shown in Figure S3. We agree with the reviewer's opinion that other sources of halogen may have an important impact on some VOCs species. However, the mean concentration of $Cl_2$ in Heshan was only 2.0±3.7 ppt during the campaign, and the average mixing ratios of $BrCl$ and $Br_2$ were less than 1.0 ppt. Since we focus on the topic on $ClNO_2$ chemistry and did not well calibrate the measurement sensitivity of $Cl_2$, $BrCl$, $Br_2$, thus we did not add the content in the revised manuscript.

[Figure]

**Figure S3.** Time series of $Cl_2$, $BrCl$ and $Br_2$.

*Ref:*

*Lopez-Hilfiker, F. D., et al., (2016), Constraining the sensitivity of iodide adduct chemical ionization mass spectrometry to multifunctional organic molecules using the collision limit and thermodynamic stability of iodide ion adducts, Atmos. Meas. Tech., 9, 1505–1512, doi:10.5194/amt-9-1505-2016.*

*Heinritzi, M., et al., (2016), Characterization of the mass-dependent transmission efficiency of a CIMS, Atmos. Meas. Tech., 9(4), 1449–1460, doi:10.5194/amt-9-1449-2016.*

7.  As mentioned above, the conclusion part can be more concise.

    We shortened the conclusion part accordingly; More details can be found in our revised manuscript.

---

## Author Comment (AC2)

Response to Editors and Reviewers

We appreciate the reviewers for their careful reading and constructive comments on our manuscript. As detailed below, the reviewer's comments are shown in black, our response to the comments is in blue. New or modified text is in red.

All the line numbers refer to Manuscript ID: acp-2022-408.

**Referee 2**

This work conducted continuous field measurements of $ClNO_2$ and $N_2O_5$ and performed comprehensive evaluation on the $ClNO_2$ chemistry as well as its contributions to radical and ozone formation under different transport pathways. The results highlight the $N_2O_5$-uptake-limited $ClNO_2$ formation and overall low contributions to $RO_2$ radical and $O_3$ formation in autumn in South China. The manuscript is generally well written with clear logic, deep analysis, and full discussion. It can be considered to accept after addressing the following minor comments.

Thanks for your positive and constructive comments.

Specific comments:

1. Line 76-77, 78-79, etc., blanks are missed in the middle of different citations. The same suggestion is given to other parts of the main text.

   Corrected throughout the manuscript.

2. Line 87-88, the sentence requires modification. e.g., "the challenge to accurately predict $ClNO_2$ and particulate nitration production".

   Here we modified as:

   Line 87-89. These gaps in the parameterization of $N_2O_5$ uptake coefficients and $ClNO_2$ yield result in the challenge of accurately predicting $ClNO_2$ and particulate nitrate production.

3. Line 148, "seldom disturbs the sampling" can be written as "to have little influence on the sampling".

   Revised accordingly.

4. Line 151, it's better to add "sometimes" after the word "are".

   Revised accordingly.

5. Line 157, add "approximately" after the word "was".

   Changed accordingly.

6. Line 203, with and without the constrains of the observed $ClNO_2$, or with and without taking $ClNO_2$ as the source of Cl radicals?

   Yes, we revised it as:

   Line 207. The impact of $O_3$ by $ClNO_2$ chemistry was assessed by differing the scenario with and without taking $ClNO_2$ as the source of Cl radicals in the model simulation.

7. Line 204-205, this operation will lead to overestimation on the contributions from $ClNO_2$ chemistry. The potential uncertainty should be described here or somewhere else.

   Since the reaction rate constant is the upper limit, it does overestimate the contribution of $ClNO_2$ chemistry. Here we added a brief discussion in the manuscript.

   Line 212. It should be noted that the setting will lead to an overestimation of the contributions from $ClNO_2$ chemistry.

8. Line 209, the average lifetime or a constant lifetime?

   Here we have rewritten it as a constant lifetime.

   Line 215. The constant lifetime corresponds to…

9. Line 210, the "was" should be "were".

   Corrected accordingly.

10. Line 209-212, this sentence is not very clear. Is there any reference to support such lifetime setting?

    Yes, this set of trace gas lifetime is consistent with our previous studies for simulating the chemistry of HOx radicals. Here we added a reference as follows.

    Line 799.

    Lu, K. D., Rohrer, F., Holland, F., Fuchs, H., Bohn, B., Brauers, T., Chang, C. C., Haseler, R., Hu, M., Kita, K., Kondo, Y., Li, X., Lou, S. R., Nehr, S., Shao, M., Zeng, L. M., Wahner, A., Zhang, Y. H., and Hofzumahaus, A.: Observation and modelling of OH and $HO_2$ concentrations in the Pearl River Delta 2006: a missing OH source in a VOC rich atmosphere, Atmospheric Chemistry and Physics, 12, 1541-1569, 10.5194/acp-12-1541-2012, 2012.

11. Line 249-250, what are the reasons for the lower abundances in 2019 than 2017? Smaller source strengths or larger sinks?

    The higher $ClNO_2$ in 2017 is due to the much higher aerosol loading with a maximum of over 400 µg m$^{-3}$, which largely promoted the conversion of $N_2O_5$ to $ClNO_2$ (namely smaller source strengths as you mentioned). While the higher $N_2O_5$ in 2017 is much more complicated since the concentration is closely related to the formation and loss of $NO_3$.

    Line 259. The difference of $ClNO_2$ level between the two campaigns conducted in 2017 and 2019 may be caused by the aerosol loading.

12. Line 258, 500 m AMSL or AGL? Are the trajectories at 100 m similar to those at 500 m?

    Here we used the AMSL and stated the manuscript. We also tested the trajectories at 100 m and 500 m and found only a small difference between them.

    Line 268. at the measurement site at 500 m AMSL height…

13. Figure 5, a RMA correlation coefficient may be better for comparing the consistent.

    We appreciate the reviewer for this suggestion. We note that other studies used the regular correlation coefficient when comparing the derived and parameterized $\gamma N_2O_5$ (e.g., Morgen et al., ACP, 2015; McDuffie et al., JGR, 2018). To keep in line with these studies, we did not used the RMA correlation coefficient, although RMA may be better here. Thus no change made with respect to this suggestion.

    *Ref:*

    *McDuffie, E. E., Fibiger, D. L., Dube, W. P., Lopez-Hilfiker, F., Lee, B. H., Thornton, J. A., Shah, V., Jaegle, L., Guo, H. Y., Weber, R. J., Reeves, J. M., Weinheimer, A. J., Schroder, J. C., Campuzano-Jost, P., Jimenez, J. L., Dibb, J. E., Veres, P., Ebben, C., Sparks, T. L., Wooldridge, P. J., Cohen, R. C., Hornbrook, R. S., Apel, E. C., Campos, T., Hall, S. R., Ullmann, K., and Brown, S. S.: Heterogeneous N2O5 Uptake During Winter: Aircraft Measurements During the 2015 WINTER Campaign and Critical Evaluation of Current Parameterizations, Journal of Geophysical Research-Atmospheres, 123, 4345-4372, 10.1002/2018jd028336, 2018.*

    *Morgan, W. T., Ouyang, B., Allan, J. D., Aruffo, E., Di Carlo, P., Kennedy, O. J., Lowe, D., Flynn, M. J., Rosenberg, P. D., Williams, P. I., Jones, R., McFiggans, G. B., and Coe, H.: Influence of aerosol chemical composition on N2O5 uptake: airborne regional measurements in northwestern Europe, Atmospheric Chemistry and Physics, 15, 973-990, DOI 10.5194/acp-15-973-2015, 2015.*

14. Line 401, add "in this study" before "than".

    Added accordingly.

15. Figure 6 and Figure 7, suggest indicating the p values of the linear correlations.

    Added accordingly.

16. Line 468, 470, 473, "power plants emissions" should be coal-fired power plant emissions.

    Revised accordingly.

17. Line 618, what does the "AH" mean? Double-check the unit mmol/mmol.

    Here the AH means absolute humidity. We added the explanation in the figure legend as "AH (absolute humidity)".

    Line 657. The unit is a typo. We corrected it as "mmol mol$^{-1}$".